# Characterising water vapour concentration dependence of commercial cavity ring-down spectrometers for continuous onsite atmospheric water vapour isotope measurements in the tropics

Shujiro Komiya[1], Fumiyoshi Kondo[2], Heiko Moossen[1], Thomas Seifert[1], Uwe Schultz[1], Heike Geilmann[1], David Walter[1,3], and Jost V. Lavric[1]

[1]Max Planck Institute for Biogeochemistry, Jena, 07745, Germany
[2]Japan Coast Guard Academy, Kure, 737-8512, Japan
[3]Max Planck Institute for Chemistry, Mainz, 55128, Germany

* *Correspondence to:* Shujiro Komiya (skomiya@bgc-jena.mpg.de)

**Abstract.** The recent development and improvement of commercial laser-based spectrometers have expanded in situ continuous observations of water vapour ($H_2O$) stable isotope compositions (e.g., $\delta^{18}O$, $\delta^2H$, etc.) in a variety of sites worldwide. However, we still lack continuous observations in the Amazon, a region that significantly influences atmospheric and hydrological cycles on local to global scales. In order to achieve accurate on-site observations, commercial water isotope analysers require regular in situ calibration, which includes the correction of $H_2O$ concentration dependence ([$H_2O$]-dependence) of isotopic measurements. Past studies have assessed the [$H_2O$]-dependence for air with $H_2O$ concentrations up to 35,000 ppm, a value that is frequently surpassed in tropical rainforest settings like the central Amazon where we plan continuous observations. Here we investigated the performance of two commercial analysers (L1102i and L2130i models, Picarro, Inc., USA) for measuring $\delta^{18}O$ and $\delta^2H$ in atmospheric moisture at four different $H_2O$ levels from 21,500 to 41,000 ppm. These $H_2O$ levels were created by a custom-built calibration unit designed for regular in situ calibration. Measurements on the newer analyser model (L2130i) had better precision for $\delta^{18}O$ and $\delta^2H$ and demonstrated less influence of $H_2O$ concentration on the measurement accuracy at each concentration level compared to the older L1102i. Based on our findings, we identified the most appropriate calibration strategy for [$H_2O$]-dependence, adapted to our calibration system. The best strategy required conducting a two-point calibration with four different $H_2O$ concentration levels, carried out at the beginning and end of the calibration interval. The smallest uncertainties in calibrating [$H_2O$]-dependence of isotopic accuracy of the two analysers were achieved using a linear-surface fitting method and a 28 h calibration interval, except for the $\delta^{18}O$ accuracy of the L1102i analyser for which the cubic fitting method gave best results. The uncertainties in [$H_2O$]-dependence calibration did not show any significant difference using calibration intervals from 28 h up to 196 h; this suggested that one [$H_2O$]-dependence calibration per week for the L2130i and L1102i analysers is sufficient. This study shows that the CRDS analysers, appropriately calibrated for [$H_2O$]-dependence, allow the detection of natural signals of stable water vapour isotopes at very high humidity levels, which has promising implications for water cycle studies in areas like the central Amazon rainforest and other tropical regions.

## 1 Introduction

Ongoing climate change has affected various aspects of global and local climate, including the hydrological cycle (Intergovernmental Panel on Climate Change, 2014). Further and more detailed understanding on how climate change affects the atmospheric hydrological system is required. Water vapour isotope compositions (e.g., $\delta^{18}O$, $\delta^2H$, $\delta^{17}O$) have been used in meteorology and hydrology to disentangle the water vapour transport, mixing and phase changes such as evaporation and condensation that govern processes of the atmospheric hydrological cycle (Dansgaard, 1964; Craig and Gordon, 1965; Galewsky et al., 2016). Incorporating water vapour isotopic information into global and regional circulation models has also

improved our understanding of how stable water isotopes are transported in the atmosphere and affected by phase changes in- and below-clouds, and how they behave in different situation of surface-atmosphere interactions (Risi et al., 2010; Werner et al., 2011; Pfahl et al., 2012). The increase in field observation of water vapour isotope compositions therefore is expected to improve our process understanding and thereby models simulating the interactions between the atmospheric
hydrological system and global climate change.

Until around 10-15 years ago, *in-situ* water vapour isotope measurements were limited due to the laborious and error-prone sampling techniques using cryogenic traps, molecular sieves, vacuum flasks, etc. (Helliker and Noone, 2010). Recent development and improvement of laser-based spectrometers have made continuous water vapour isotope composition measurements at a high temporal resolution possible. The number of onsite measurements of stable water vapour isotope
compositions across the world has increased in the last decade (Wei et al., 2019). So far, there are many studies based on field water isotopic measurements available in polar and midlatitude regions, some in the subtropics (e.g., Bailey et al., 2013; Gonzalez et al., 2016) but only very few in the tropics (e.g., Tremoy et al., 2012; Aemisegger et al., 2020). Particularly studies in tropical continental regions, such as the Amazon basin region, are rare. Yet, understanding the hydrological processes in the Amazon basin is crucial as it significantly influences the atmospheric convective circulation in the tropics
and beyond (Coe et al., 2016; Galewsky et al., 2016). Thus, in-situ continuous measurements of water vapour isotope compositions in the Amazon region will improve our comprehension of the Amazonian hydro-climatological system and its interaction with global climate (Coe et al., 2016; Galewsky et al., 2016).

Recent field observations for water vapour isotopes have mainly utilized two commercial laser-based instruments: Picarro cavity ring-down spectroscopy (CRDS) and Los Gatos Research off-axis integrated cavity output spectroscopy (OA-ICOS)
analysers (Galewsky et al., 2016; Wei et al., 2019). The CRDS analysers have been used in most of the field sites that are registered in the Stable Water Vapor Isotopes Database (SWVID) website that archives onsite high-frequency water vapour isotope data (Wei et al., 2019). Globally, five Picarro CRDS models (i.e., L1102i, L1115i, L2120i, L2130i, L2140i sorted by oldest to newest) are in operation at various field sites. Aemisegger et al. (2012) demonstrated that a recent model (L2130i) has better precision and accuracy compared to an older model (L1115i) due to the improved spectroscopic fitting algorithms.
Even with improved analysers, CRDS instruments still require regular calibration (e.g., 3-24 hour frequency) (Aemisegger et al., 2012; Delattre et al., 2015; Wei et al., 2019). The main calibration issue is that the measurement quality of water vapour isotopic compositions depends on water vapour ($H_2O$) concentration (hereinafter called "[$H_2O$]-dependence"; Schmidt et al., 2010, Tremoy et al., 2011, Aemisegger et al., 2012, Bailey et al., 2015, Delattre et al., 2015). The [$H_2O$]-dependence of Picarro analysers has been assessed over a $H_2O$ concentration range spanning 200 to 35,000 ppm (Schmidt et al., 2010;
Aemisegger et al., 2012; Steen-Larsen et al., 2014; Bailey et al., 2015; Delattre et al., 2015), and only rarely above 35,000 ppm (Tremoy et al. 2011).

However, $H_2O$ concentrations within the Amazon tropical rainforest canopy (e.g., the Amazon Tall Tower Observatory (ATTO) site; see Andreae et al., 2015) exceed 35,000 ppm on a daily basis and occasionally 40,000 ppm. In addition, Moreira et al. (1997) observed the diel variation pattern in $H_2O$ concentration in the Amazon tropical rainforest was mostly
similar to that in $\delta^{18}O$ and $\delta^2H$ of water vapour. The diel relationship between $H_2O$ concentration and isotopes may lead to over- or under-estimation of isotopic values measured by CRDS analysers in the Amazon tropical rainforest. Thus, for in-situ water vapour isotope measurements by CRDS analysers in the Amazon tropical rainforest, the [$H_2O$]-dependence of CRDS analysers under high moisture conditions (> 35,000 ppm $H_2O$) needs to be assessed and corrected.

The primary aim of this study was to characterise two CRDS analysers (L1102i and L2130i) for measuring $\delta^{18}O$ and $\delta^2H$ of
water vapour in high atmospheric moisture expected at the ATTO site (~ 150 km NE of Manaus, Brazil), where we intend to conduct continuous in-situ observations. Over a two-week period, we examined the effects of $H_2O$ concentration on isotopic measurement precision and accuracy for both an old (L1102i) and a new CRDS models (L2130i). They were both connected to our custom-made calibration system that regularly supplied standard water vapour samples at four different $H_2O$

concentrations covering high moisture conditions (21,500 to 41,000 ppm). Standard water vapour samples were made from two standard waters, almost covering the previously reported isotopic ranges ($\delta^{18}$O = -19.4 to -6.7 ‰ and $\delta^2$H = -151 to -42 ‰) for water vapour samples from Manaus or from the Ducke Reserve near Manaus (Matsui et al., 1983; Moreira et al., 1997; IAEA/WMO, 2020). We also assessed which [$H_2O$]-dependence calibration strategy can best reduce measurement

uncertainty of the two CRDS models. Based on the uncertainty quantification presented, we discussed whether the CRDS analysers with our calibration setup can sufficiently detect natural signals of stable water vapour isotopes expected at the ATTO site.

## 2    Materials and Methods

### 2.1 Calibration system

A setup with a commercial vaporizer coupled with a standard delivery module (A0211 and SDM, A0101, respectively; Picarro Inc., Santa Clara, CA, USA), guarantees the delivery of standard water vapour samples up to 30,000 ppm of $H_2O$, which does not cover the $H_2O$ concentration range we are expecting for the Amazon rainforest. In addition, according to discussion with Picarro's technicians, there is no easy way to run an A0101 with the L1102i model. Therefore, we built a calibration system to routinely and automatically conduct onsite-calibration of CRDS analysers (Fig. 1). The main units of

the calibration system are a syringe-pump, a vaporizer and a dry-air supply unit (Fig. 1a). The syringe-pump (Pump 11 Pico Plus Elite, Harvard Apparatus, Holliston, MA, USA) takes 3.3 mL standard water from a 2L reservoir bag (Cali-5-Bond™, Calibrated Instruments, Inc., Ardsley, New York, USA), and delivers the standard water into a vaporizer unit with a constant water flow of 1.9 µL min⁻¹ (Figs. 1a and 1b). To maintain accuracy of the syringe-pump's infusion over a long term, two guide rods and a lead screw of the syringe pump need to be properly lubricated every 100 hours of operation (i.e., injecting

and withdrawing). The vaporizer unit that was modified from an A0211 is comprised of a heater, vaporization chamber and buffer reservoir, which is enclosed in a copper pipe and heated at 140 °C, covered by insulation material to reduce heat dissipation and help to reduce the memory effects between different water vapour isotopic measurements. Dried ambient air, recommended as a carrier gas for calibration by Aemisegger et al. (2012), was supplied into the heated vaporizer unit from a dry-air unit made up of a compressor, water separator, mist separators, membrane dryer (IDG60SAM4-F03C, SMC, Tokyo,

Japan), precision regulator (IR1000, SMC, Tokyo, Japan) and flow regulator (Figs. 1a and 1c). We chose the SMC's membrane dryer because SMC guarantees a long-term operation (e.g., 10 years or more by 10 hours/day operation) without replacing the membrane module. The dry-air unit and mass flow controller 1 (MFC1) (1179B, MKS GmbH, Munich, Germany) provide the vaporizer unit with a steady flow of dried ambient air with a dew point temperature of -32 °C or below (~300 ppm of $H_2O$ or below), operated at 50 mL min⁻¹ flow rate and 17.2-20.7 kPa flow pressure. The dry air entering the

vaporizer is heated through the heater line, speeding up the evaporation of the infused standard water inside the vaporization chamber without fractionation. Furthermore, the heated carrier gas also helps reducing the memory effect of the measurements. The subsequent standard water vapour was well mixed inside a bigger buffer reservoir compared to A0211. The customized heating system and buffer reservoir enabled us to produce a high moisture stream of standard water vapour samples, then delivered through the multiposition valve (Model EMTMA-CE, VICI Corp., Houston, TX, USA), switching

flow paths between the calibration and routine analysis mode of the two CRDS analysers: L1102i and L2130i. To minimise tubing memory effects on water vapour isotopic measurements, we connected the vaporizer unit and CRDS analysers with stainless steel tubing constantly held at 45 °C with heating tapes to avoid condensation inside the tubes (c.f., Schmidt et al., 2010, Tremoy et al., 2011). Before reaching the CRDS analysers, the transported standard water vapour was diluted with the dried ambient air via a dilution line and adjusted to an intended concentration level by regulating the dilution dry air flow

rate using MFC2. The total flow rate of both the calibration and dilution lines exceeded the suction flow rates of the two CRDS analysers (~50 mL min⁻¹ in total). The excess air was exhausted through an overflow port.

## 2.2 Water vapour concentration dependence experiment

We conducted a continuous operation of the L1102i and L2130i analysers over a two-week period in June 2019 in an air-conditioned laboratory at Max Planck Institute for Biogeochemistry (MPI-BGC, Jena, Germany). The two CRDS analysers measured water vapour ($H_2O$) concentration, $\delta^{18}O$ and $\delta^2H$ of outside/room air samples from a profile gas-stream switching system (not shown here) or $H_2O$ concentration, $\delta^{18}O$ and $\delta^2H$ of water vapour samples supplied from the calibration system (Fig. 1a). Since $H_2O$ concentration values measured by old CRDS models (e.g., L1102i) are biased due to the self-broadening effect of water vapour (Winderlich et al., 2010; Rella et al., 2013), $H_2O$ concentration measuring by the L1102i were corrected by a nonlinear calibration fitting, determined by Winderlich et al., (2010) and recommended for old CRDS models (e.g., G1301 $CO_2/CH_4/H_2O$ analyser) by Rella (2010).

We also simulated regular automated calibration operation designed for field operations over the two-week period to regularly supply the two CRDS analysers with standard water vapour samples at four different concentration levels from 21,500 to 41,000 ppm. We prepared two different working standard waters (DI1 and DI2) made of deionized water to avoid clogging the heated tubes and chamber inside the vaporizer unit with contaminants. Stable water isotope compositions ($\delta^{18}O$ and $\delta^2H$) of the DI1 and DI2 standards were analysed at the MPI-BGC stable isotope laboratory (BGC-IsoLab) using Isotope Ratio Mass Spectrometry (IRMS). For details on the IRMS analysis, we refer readers to Gehre et al. (2004). The DI1 and DI2 standards were calibrated against VSMOW and SLAP via in-house standards: DI1-$\delta^{18}O$ = -25.07 ± 0.16 ‰, DI1-$\delta^2H$ = -144.66 ± 0.60 ‰, DI2-$\delta^{18}O$ = -3.69 ± 0.15 ‰, DI2-$\delta^2H$ = -34.30 ± 1.00 ‰ (also see section S1 in the Supplement). The isotopic span of the DI1 and DI2 almost covers the previously reported range of $\delta^{18}O$ (-19.4 to -6.7 ‰) and $\delta^2H$ (-151 to -42 ‰) for water vapour samples in the Ducke Reserve near Manaus or in Manaus, located near the ATTO site (Matsui et al., 1983; Moreira et al., 1997; IAEA/WMO, 2020). For calibration, we alternated between the two standard waters. One calibration run required 75 minutes, of which the first 30 minutes were used for stabilizing the produced standard water vapours at the highest concentration level and delivering it to the CRDS analysers. Subsequently the calibration system created stepwise lower concentration levels of the standard water vapour every 15 minutes by regulating the dilution flow rate. One calibration run consisted of a 4-point concentration calibration at approximately 41,000 ppm, 36,000 ppm, 29,000 ppm, and 21,500 ppm. The actual measured mean and standard deviation of $H_2O$ concentration at the respective four concentration level for all the calibration cycles during the two-week operation are shown in Table 1. The dilution flow rates for the different concentration levels were set to 9 sccm (41,000 ppm), 14 sccm (36,000 ppm), 21 sccm (29,000 ppm) and 28 sccm for 21,500 ppm. The set-point values were not changed during the 2-week test, thus simulating remote automatic onsite calibration runs. We used the last 7-minutes of data collected at each concentration level for the calibration assessment of the CRDS analysers. Immediately after a calibration cycle, the syringe-pump drained the remaining standard water inside the tube between the vaporization chamber and 3-way solenoid valve 6 (SV6) through the waste line and then washed the inner space between the vaporization chamber and the SV1 valve 3 times with the standard water scheduled for the next calibration cycle (Figs. 1a and 1b). Subsequently, the rinsed vaporization chamber was fully dried with air from the dry-air unit for 2 to 4 hours. These rinsing and drying steps were introduced to minimize the residual memory effect from the last calibration cycle on standard water vapour isotopic compositions during the next calibration run. We started the next calibration cycle seven hours after the start time of the last calibration cycle (i.e., the interval of the same working standard was 14 hours). Throughout the entire experimental period, the calibration system conducted automatically 24 calibration runs for each standard water, which used 160 mL each standard water in total.

We calculated isotopic deviations at each concentration level for each calibration cycle to evaluate [$H_2O$]-dependence on isotopic measurement accuracy. The isotopic deviations at each concentration level were obtained from the difference between measured isotopic values at each concentration level during each calibration cycle and assigned reference values at

21,500 ppm on each calibration. We selected the 21,500 ppm level as the reference $H_2O$ condition because Picarro guarantees high $\delta^{18}O$ and $\delta^2H$ precision of CRDS analysers between 17,000 – 23,000 ppm of $H_2O$, and to make the results comparable with a similar past study (Tremoy et al., 2011) that assigned 20,000 ppm as the reference level. The assignment of reference values for each calibration cycle enabled us to assess isotopic biases that were mainly due to [$H_2O$]-dependence but not for other effects (e.g., drift effects on $\delta^{18}O$ and $\delta^2H$ accuracy between each calibration cycle).

## 2.3 Calibration for water vapour concentration dependence

We devised four strategies, referred to here as DI1, DI2, DI1-DI2*1Pair, DI1-2*2Pairs, to use the automated calibration system to determine and correct for [$H_2O$]-dependence, and used the two-week operation to assess which calibration strategy decreased the uncertainties in $\delta^{18}O$ and $\delta^2H$ measurements the most. The DI1 and DI2 calibration strategies used a single standard water (DI1 or DI2) to correct for [$H_2O$]-dependence. In contrast, the DI1-DI2*1Pair and DI1-2*2Pairs strategies used the two standard waters (DI1 and DI2) for calibrating [$H_2O$]-dependence because we considered that [$H_2O$]-dependence might change between the two standard waters according to recent studies (Bonne et al., 2019; Weng et al., 2020). In addition, the DI1-2*2Pairs strategy used more calibration data than the DI1-DI2*1Pair strategy to obtain more robust calibration fittings of [$H_2O$]-dependence.

Figure 2 summarises the overview of the four calibration strategies. DI1 and DI2 refer to the two standard waters, measured at the four different concentration levels, and an identity number is assigned to the respective 48 calibration cycles (ID: 1 − 48), ordered by time during the experimental period (Fig. 2a). Hereinafter, we explain how to obtain calibration fittings and assess uncertainties of obtained calibration fittings by using the DI1-2*2Pairs strategy as an example. The DI1-2*2Pairs strategy uses two pairs of DI1 and DI2 calibration cycles (e.g., ID-[3,4] & [7,8] at 28 h interval) for obtaining calibration fittings of [$H_2O$]-dependence at intervals from 28 h to 196 h (Fig. 2b). The DI1-2*2Pairs strategy utilized four two-dimensional (2D) and a three-dimensional (3D) fitting methods (i.e., linear, quadratic, cubic, quartic, and linear-surface fitting methods) to obtain [$H_2O$]-dependence calibration fitting parameters. For instance, the DI1-2*2Pairs strategy at 28 h interval with ID-[3,4] & [7,8] obtained total 5 [$H_2O$]-dependence calibration fittings for each isotope accuracy.

As an example, Figure 3 illustrates five calibration fittings for [$H_2O$]-dependence of $\delta^{18}O$ accuracy for each CRDS analyser, acquired from two pairs of DI1 and DI2 calibration cycles (e.g., 3,4 and 7,8) following the DI1-2*2Pairs strategy at a 28 h interval (also c.f., Fig. 2b). The respective 2D fitting is acquired from a relationship between $H_2O$ concentrations, measured at 4 different levels, and $\delta^{18}O$ deviation at each of the 4-point levels (the blue dots in Figs. 3a-h). The linear-surface fitting also involves measuring the $\delta^{18}O$ value at each concentration level as an independent variable (Figs. 3i and 3j). The procedure for calculating isotopic deviations at each concentration level was the same as described in section 2.2.

The quantitative evaluation of uncertainties in [$H_2O$]-dependence calibration was conducted by means of root mean square error (RMSE) between actual observed and predicted isotopic deviation values by obtained [$H_2O$]-dependence fittings. We calculated RMSE value for each of the [$H_2O$]-dependence 2D or 3D fittings as follows:

$$RMSE = \sqrt{\frac{1}{n}\sum_{i=1}^{n}\left(\delta_{(obs)} - \delta_{(pred)}\right)^2} \qquad (1)$$

where $\delta_{(obs)}$ is the actual observed deviation value of $\delta^{18}O$ or $\delta^2H$ at each concentration level from measurement cycles, $\delta_{(pred)}$ is the predicted deviation value of $\delta^{18}O$ or $\delta^2H$ at each concentration level, and n is the sample number. The measurement cycles represent calibration cycles that were not used for estimating [$H_2O$]-dependence, over an interval period between one or two calibration pairs. For example, in the case of the DI1-2*2Pairs strategy at 28 h interval with ID-[3,4] & [7,8], a DI1 and D2 calibration cycles (i.e., DI1: 5, DI2: 6) are regarded as measurement cycles (Fig. 2). The measured $\delta^{18}O$ deviation value at each concentration level during the measurement cycles (e.g. red dots: DI1, ID-5; green dots: DI2, ID-6; Fig. 3) represents the value of $\delta_{(obs)}$ at each concentration level. The predicted isotopic deviation value ($\delta_{(pred)}$) at each concentration

level during the measurement cycles (e.g., DI1: 5, DI2: 6) is calculated by each of the $[H_2O]$-dependence 2D or 3D fittings, applied to measured value of $H_2O$ concentration at each concentration level (the open triangles in Figs. 3a-3h) or to measured values of both $H_2O$ concentration and $\delta^{18}O$ at each concentration level during the measurement cycles (the open triangles in Figs. 3i and 3j). The example of calculated $\delta^{18}O$ RMSE of each fitting method for each CRDS analyser is shown on each plot in Fig. 3.

We conducted $\delta^{18}O$ and $\delta^{2}H$ RMSE evaluation for all the $[H_2O]$-dependence fittings that were obtained from all the respective two pairs of DI1 and DI2 calibration cycles at each interval period (28 - 196 h) following the DI1-2*2Pairs strategy over the entire two-week period (c.f., Fig. 2b). For instance, since the DI1-2*2Pairs strategy at 28 h interval formed 43 two pairs in total (i.e., [1,2]-[5,6], [2,3]-[6,7], [3,4]-[7,8] ~ [42,43]-[46,47], [43,44]-[47,48]) (c.f., Fig. 2b), we calculated $\delta^{18}O$ and $\delta^{2}H$ RMSE values for each of 43 $[H_2O]$-dependence linear, quadratic, cubic, quartic, or linear-surface fittings.

Compared with the DI1-2*2Pairs strategy, the DI1, DI2, and DI1-DI2*1Pair calibration strategies used a pair of only DI1 calibrations, a pair of only DI2 calibrations, and a pair of DI1 and DI2 calibrations, respectively, for acquiring $[H_2O]$-dependence fittings at intervals from 14 h to 196 h (DI1, DI2 calibration strategies) or from 21 h to 203 h (DI1-DI2*1Pair calibration strategy) (Fig. 2b). The DI1 and DI2 calibration strategies utilized the four 2D fitting methods for obtaining $[H_2O]$-dependence calibration fittings, whereas the DI1-DI2*1Pair calibration strategy used the four 2D and one 3D fitting methods as with the DI1-2*2Pairs strategy. For each calibration strategy, we also calculated RMSE value for each of the $[H_2O]$-dependence 2D or 3D fittings.

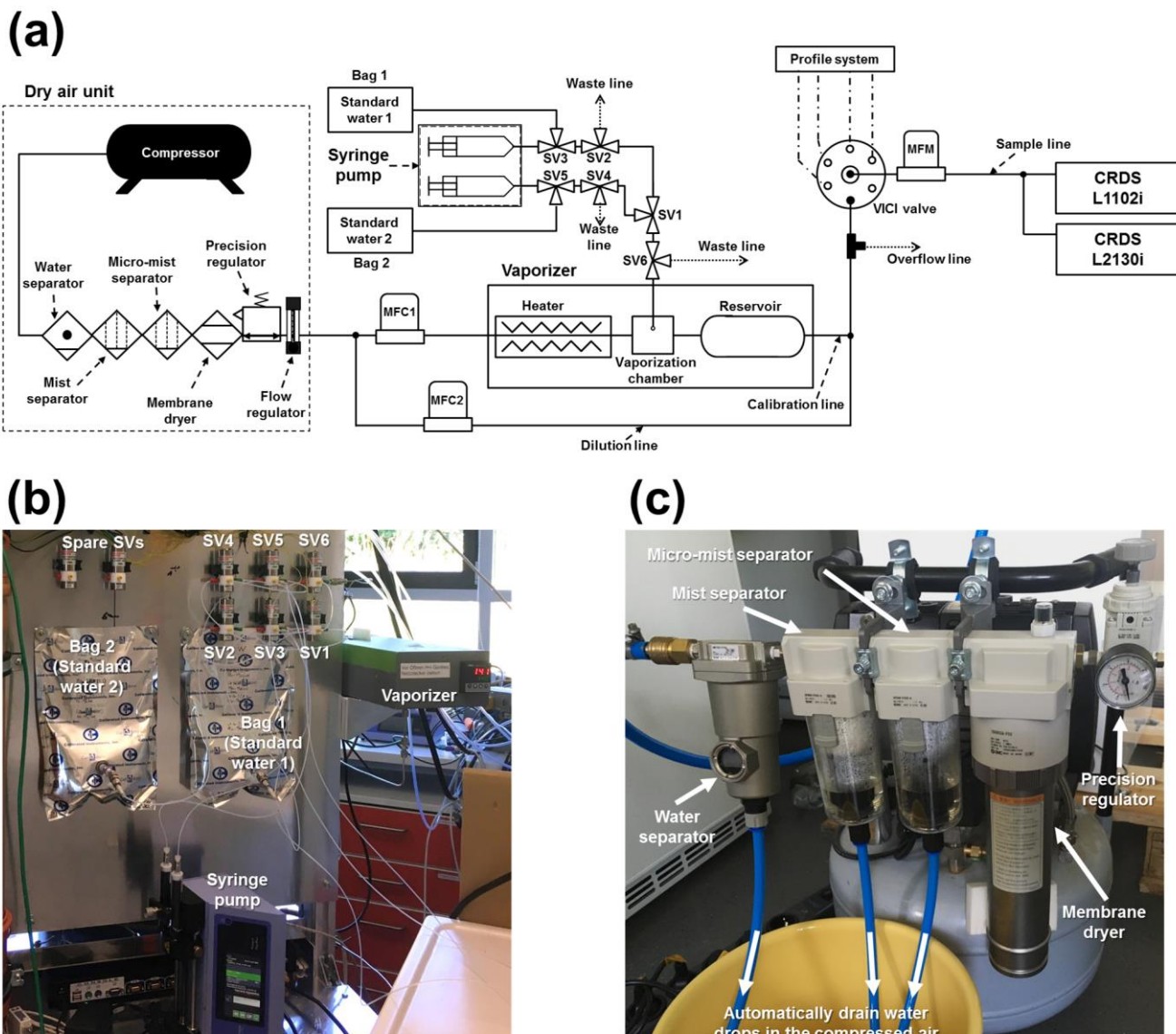

*Figure 1* (a) Schematic diagram of the calibration system, including the dry air unit. MFC, MFM and SV denote mass flow controller, mass flow meter, and 3-way solenoid valve, respectively. The profile system, prepared for in situ observation at the Amazon Tall Tower Observatory site (c.f., Andreae et al. 2015) in the Amazon tropical forest, is not described in this article. The diagram is not to scale. Panel (b) shows the photo of the main part of the calibration system: a vaporizer, syringe pump, 2L reservoir bags, and solenoid valves. Panel (c) shows the photo of the main part of the dry air unit: a water separator, mist separators, membrane dryer, precision regulator, and drain lines for water drops in the compressed air.

**Table 1** Standard deviations of $\delta^{18}O$ and $\delta^{2}H$ at each concentration level from all 24 calibration runs for the respective DI1 and DI2 standard waters. The standard deviations were calculated from all 24 data set of raw 7-min average values, obtained by L2130i and L1102i on each calibration run. The mean values of $H_2O$ concentration at each concentration level were also calculated from all 24 data set of raw 7-min average values, obtained by L2130i and L1102i on each calibration run.

| Picarro | Standard water | | | | | |
| | DI1 (n=24) | | | DI2 (n=24) | | |
| | $H_2O$ concentration (ppm) | $\delta^{18}O$ (‰) | $\delta^{2}H$ (‰) | $H_2O$ concentration (ppm) | $\delta^{18}O$ (‰) | $\delta^{2}H$ (‰) |
|---|---|---|---|---|---|---|
| L2130i | 21656.2 ± 243.6 | 0.09 | 0.45 | 21289.8 ± 220.0 | 0.08 | 0.55 |
| | 29144.1 ± 578.6 | 0.08 | 0.35 | 28473.0 ± 514.7 | 0.11 | 0.59 |
| | 36593.8 ± 676.2 | 0.13 | 0.68 | 35701.3 ± 565.7 | 0.12 | 0.68 |
| | 41803.0 ± 653.9 | 0.13 | 0.68 | 40744.4 ± 543.1 | 0.12 | 0.71 |
| L1102i | 20967.1 ± 253.5 | 0.14 | 1.01 | 20563.2 ± 244.9 | 0.11 | 0.99 |
| | 28957.3 ± 634.2 | 0.12 | 0.72 | 28244.0 ± 545.4 | 0.15 | 1.08 |
| | 37250.0 ± 756.5 | 0.18 | 0.90 | 36246.2 ± 690.1 | 0.20 | 1.06 |
| | 43387.8 ± 759.8 | 0.23 | 0.79 | 42172.1 ± 695.9 | 0.21 | 0.95 |

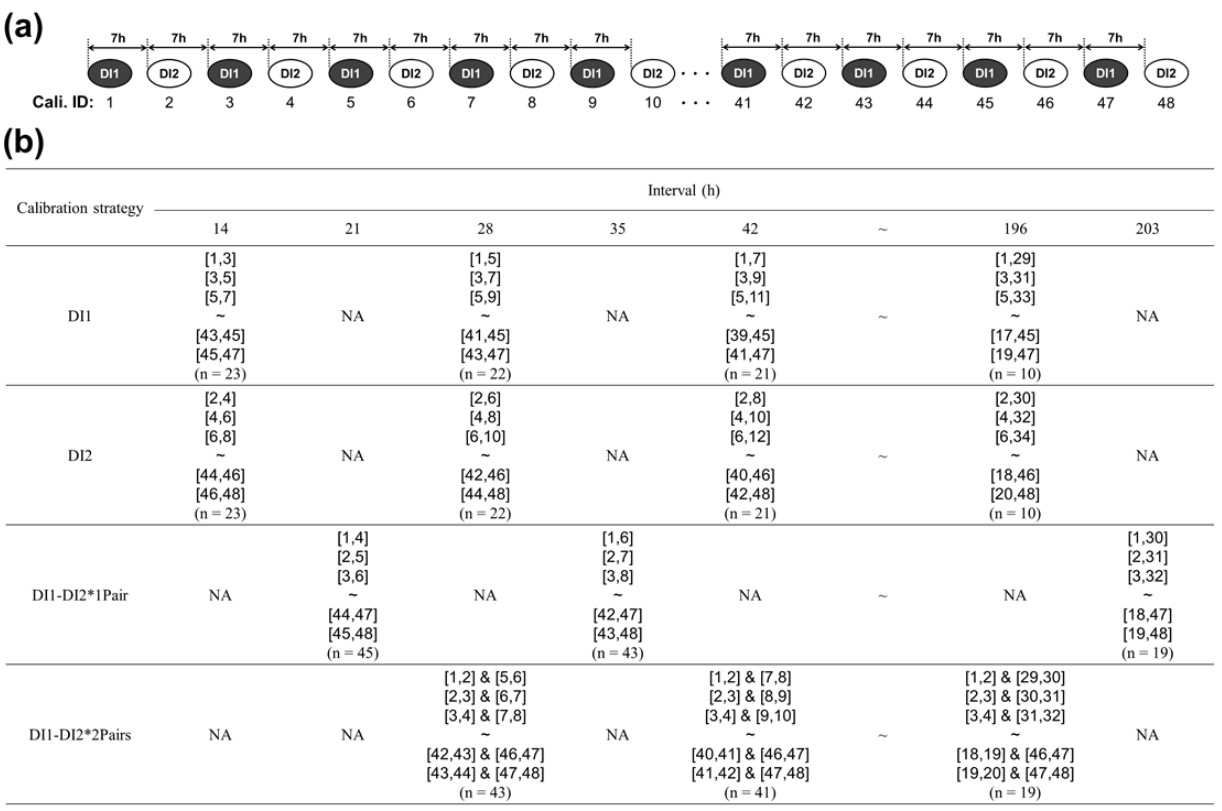

*Figure 2* Overview of the four different [$H_2O$]-dependence calibration strategies (DI1, DI2, DI1-DI2*1Pair, DI1-2*2Pairs) for assessing [$H_2O$]-dependence uncertainties of the L2130i and L1102i analysers related to the length of the calibration interval. (a) The top diagram shows a part of the total 48 calibration cycles, including both the DI1 and DI2 standard waters, with 7-hour interval and identity number (ID: 1 − 48) as an example. (b) The bottom table presents a part of which calibration cycles the four calibration strategies utilize for obtaining [$H_2O$]-dependence calibration fittings at different intervals as an example. The respective DI1 and DI2 strategies used a pair of only DI1 calibration cycles and a pair of only DI2 calibration cycles for obtaining calibration fittings of [$H_2O$]-dependence at intervals from 14 h to 196 h. For example, the DI1 calibration strategy at 14 h interval with ID-[1,3] used data sets from a pair of DI1 calibration cycles (ID-1 and ID-3) to obtain [$H_2O$]-dependence calibration fittings. The DI1-DI2*1Pair and DI1-2*2Pairs strategies used a pair of DI1 and DI2 calibration cycles, and two pairs of DI1 and DI2 calibration cycles, respectively, for obtaining calibration fittings of

[H$_2$O]-dependence at intervals from 21 h to 203 h (DI1-DI2*1Pair strategy) or from 28 h to 196 h (DI1-DI2*2Pairs strategy). At each interval, the respective calibration strategy made all the pairs of each set calibration cycles (e.g., 43 two pairs at 28 h interval for the DI1-DI2*2Pairs strategy), and then obtained calibration fittings from each pair of calibration cycles. For instance, as the DI1-DI2*2Pairs strategy utilized four two-dimensional (2D) and a three-dimensional (3D) fitting methods (i.e., linear, quadratic, cubic, quartic, and linear-surface fitting methods) (c.f., section 2.3), the DI1-DI2*2Pairs strategy at 28 h interval obtained 215 calibration fittings in total (= 5 fitting methods × 43 two pairs) for the respective $\delta^{18}$O and $\delta^2$H accuracy. The procedure for assessing uncertainties in the obtained [H$_2$O]-dependence calibration fittings is described in section 2.3.

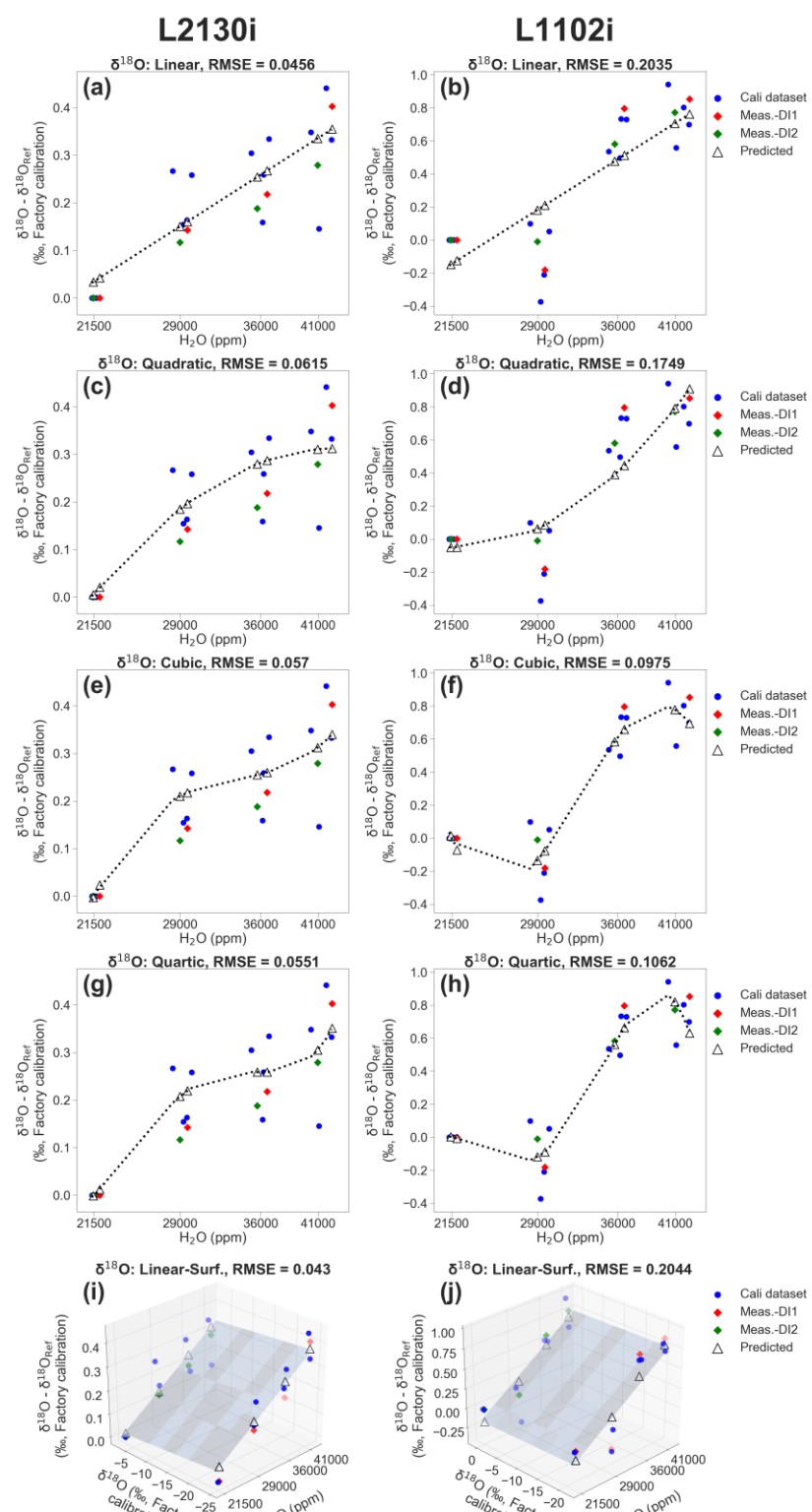

*Figure 3* Example of calibrating [$H_2O$]-dependence of $\delta^{18}O$ accuracy for the respective L2130i and L1102i analysers by using four two-dimensional (2D) and a three-dimensional (3D) fitting methods (i.e., linear, quadratic, cubic, quartic, and linear-surface fitting methods) according to the DI1-2*2Pairs calibration strategy at 28 h interval with ID-[3,4] & [7,8] (c.f., Fig. 2b). Calibrating (a) L2130i's and (b) L1102i's [$H_2O$]-dependence by using the linear fitting method. Calibrating (c) L2130i's and (d) L1102i's [$H_2O$]-dependence by using the quadratic fitting method. Calibrating (e) L2130i's and (f) L1102i's [$H_2O$]-dependence by using the cubic fitting method. Calibrating (g) L2130i's and (h) L1102i's [$H_2O$]-dependence by using the quartic fitting method. Calibrating (i) L2130i's and (j) L1102i's [$H_2O$]-dependence by using the linear-surface fitting method. The blue dots on each plot represent data sets from two pairs of DI1 and DI2 calibration cycles (i.e., [3,4] & [7,8]) for obtaining each of the five calibration fittings. The respective 2D [$H_2O$]-dependence calibration fitting derives from

a relationship between measured $H_2O$ concentration and $\delta^{18}O$ deviation, equivalent to a difference between a measurement value of $\delta^{18}O$ at each of four concentration levels and that at 21,500 ppm level ($\delta^{18}O_{Ref}$) on each calibration (a-h). The 3D [$H_2O$]-dependence calibration fitting also involves a measurement value of $\delta^{18}O$ value at each concentration level as an independent variable (i-j). The red and green diamonds denote the actual observed $\delta^{18}O$ deviations from unused DI1 (ID-5) and DI2 (ID-6) calibration cycles, respectively. The predicted $\delta^{18}O$ deviations, denoted as open triangles, on each plot were calculated from each [$H_2O$]-dependence calibration fitting, applied to measured $H_2O$ concentrations (2D fitting methods; a-h) or to both measured $H_2O$ concentration and $\delta^{18}O$ (3D fitting method; i-j) during the unused calibration cycles. The RMSE value on each plot was calculated from a difference between actual observed and predicted deviation values of $\delta^{18}O$.

## 3    Results and Discussions

### 3.1    Measurement precision over the two-week period

Figure 4 shows example times series of 7-min mean $H_2O$ concentration, $\delta^{18}O$, and $\delta^2H$, calculated from raw L2130i measurement data for the highest and lowest $H_2O$ concentrations (i.e., ~41,000 and ~21,500 ppm) over the entire DI1 standard water calibration runs for the two week period. The precision is defined as the standard deviation (σ) of all the raw 7-min average values over the 24 calibrations. The temporal changes in measured $H_2O$ concentration at 21,500 ppm varied with $H_2O$-σ = 243.6 ppm (1.1 %) over the whole period, whereas the highest moisture condition (~41,000 ppm) had larger variation of $H_2O$ concentration ($H_2O$-σ: 653.9 ppm or ~1.5 %; Figs. 4a and 4b). The larger variability at 41,000 ppm likely derived from difficulty in establishing and delivering a stable high moisture stream from the calibration unit to L2130i, and possibly residual memory effects inside the tube line and L2130's measurement cell even with the well-heated condition (cell temperature = 80 °C) due to the extremely high moisture. One solution of the possible residual memory effects would be an increase in the tube-heater's temperature above 45 °C. In addition, the larger variation in $H_2O$ concentration at 41,000 ppm may have been influenced by instability of the calibration system and a saturation effect inside the cavity measuring $H_2O$ concentration near the upper limit (50,000 ppm). Moreover, the less stable $H_2O$ signal at the highest concentration level results in lower precisions of $\delta^{18}O$ (σ = 0.13 ‰) and $\delta^2H$ (σ = 0.68 ‰) compared with at the lowest concentration level ($\delta^{18}O$-σ = 0.09 ‰ and $\delta^2H$-σ = 0.45 ‰) (Figs. 4c-f).

Table 1 summarizes the precision of $\delta^{18}O$ and $\delta^2H$ for each concentration level on each standard water for the L2130i and L1102i analysers. For both standard waters, the L2130i analyser had higher $\delta^{18}O$ and $\delta^2H$ precision than the L1102i analyser, likely due to the improved fitting algorithm used for the L2130i analyser (Aemisegger et al., 2012). In addition, the L2130i analyser had higher $\delta^{18}O$ (σ ≤ 0.11 ‰) and $\delta^2H$ (σ ≤ 0.59 ‰) precision under 30,000 ppm than over 30,000 ppm: $\delta^{18}O$ -σ ≥ 0.12 ‰ and $\delta^2H$ -σ ≥ 0.68 ‰ (Table 1). The L1102i analyser also had higher precision below 30,000 ppm relative to over 30,000 ppm except $\delta^2H$ precision for DI2 (Table 1). These findings indicate that both analysers can measure stable water isotopes more precisely for water vapour samples below 30,000 ppm.

Across concentration levels for DI2, both analysers had the highest $\delta^{18}O$ and $\delta^2H$ measurement precision at 21,500 ppm level with the lowest $H_2O$ variation ($H_2O$-σ ≤ 244.9 ppm) except $\delta^2H$ precision of L1102i. In contrast, for DI1, both analysers had the highest $\delta^{18}O$ and $\delta^2H$ measurement precision at 29,000 ppm even though variability in $H_2O$ concentration measurement was higher at 29,000 ppm ($H_2O$-σ ≥ 578.6 ppm) than at 21,500 ppm ($H_2O$-σ ≤ 253.5 ppm). The high measurement precision of L2130i and L1102i even with the larger $H_2O$ concentration variability indicates that the measurement precision of the L2130i and L1102i analysers was not largely influenced by the instability of the calibration unit. Furthermore, the L1102i's $\delta^{18}O$ and $\delta^2H$ precision at 21,500 ppm ($\delta^{18}O$-σ = 0.11-0.14 ‰ and $\delta^2H$-σ = 0.99-1.01 ‰) were similar or better than those reported by Delattre et al. (2015) at 20,000 ppm for the L1102i ($\delta^{18}O$ and $\delta^2H$ precision of 0.08-0.19 ‰ and 1.5-2.0 ‰

respectively based on 40 calibration data over 35 days). This proves that the calibration system has a negligible effect on the isotopic measurement precision for L2130i and L1102i analysers.

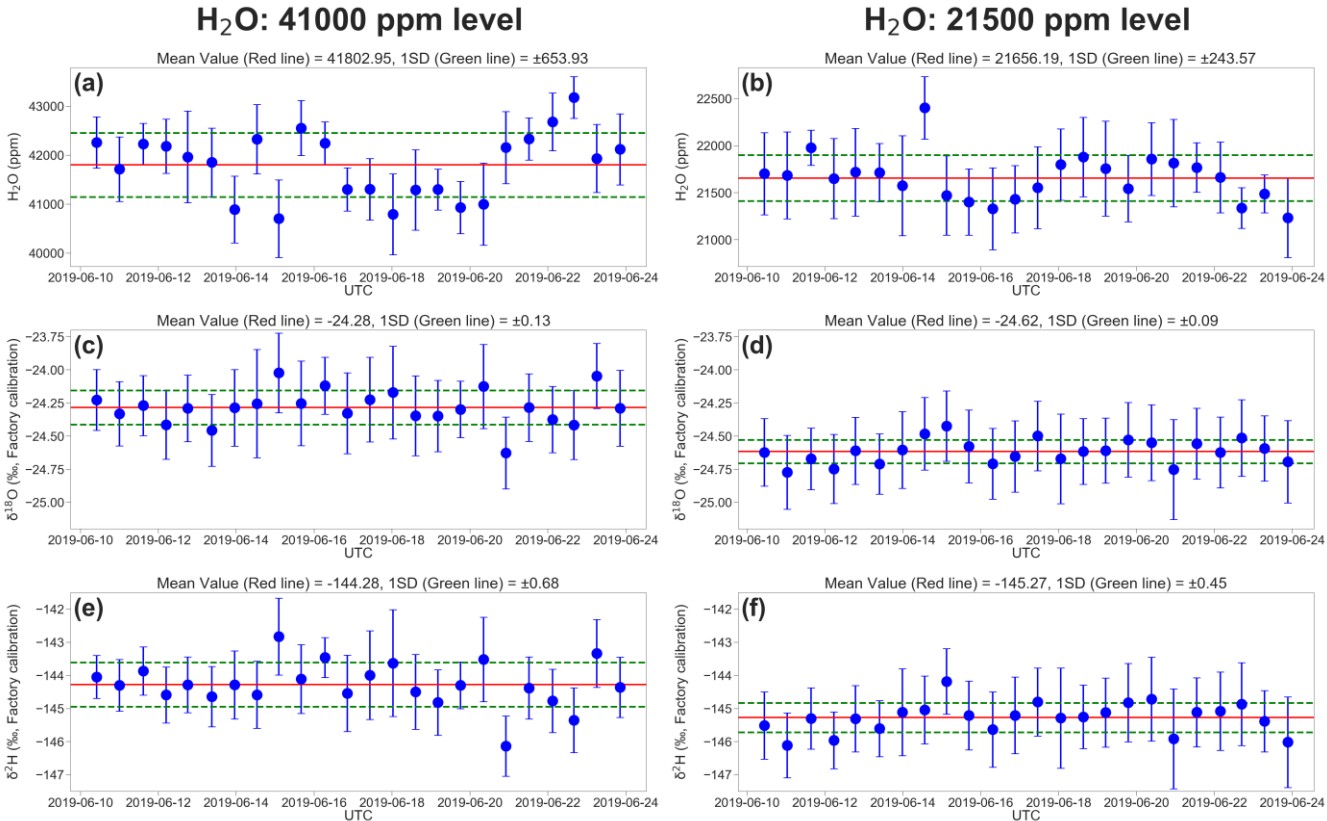

*Figure 4* Two-week evolution of the L2130i's measurements for the DI1 standard water in (a) $H_2O$ concentration at 41,000 ppm level, (b) $H_2O$ concentration at 21,500 ppm level, (c) $\delta^{18}O$ at 41,000 ppm level, (d) $\delta^{18}O$ at 21,500 ppm level, (e) $\delta^2H$ at 41,000 ppm level, and (f) $\delta^2H$ at 21,500 ppm level. Each value is a 7-minute average of raw measurement data from the L2130i, and error bars are one standard deviation of 7-minutes. The red and green lines show the average and standard deviation through the 2-week measurement period.

## 3.2 Accuracy of isotope values for water vapour concentration dependence

For the L2130i analyser, the $\delta^{18}O$ deviation of both standard waters from reference values at 21,500 ppm gradually increased with $H_2O$ concentration, reaching a maximum median value of 0.32 ‰ for DI1 and of 0.28 ‰ for DI2 (Fig. 5a). For $\delta^{18}O$, these differences were significant for both standard waters between 41,000 and 36,000 or 29,000 ppm (Fig. 5a, Welch's t-test, p<0.01), but differences between 36,000 and 29,000 ppm were not significant (Fig. 5a, Welch's t-test, p>0.09). As with $\delta^{18}O$, the values of $\delta^2H$ measured with the L2130i for both standard waters differed significantly between 41,000 ppm and the lower $H_2O$ concentrations (Fig. 5b, Welch's t-test, p<0.05), without a significant difference between 29,000 ppm and 36,000 ppm (Fig. 5b, Welch's t-test, p>0.86). These results indicate accurate measurement of both $\delta^{18}O$ and $\delta^2H$ using the L2130i analyser require correction for [$H_2O$]-dependence under high moisture conditions (> 36,000 ppm $H_2O$).

The differences in the L2130i's deviation of each isotope from reference values at 21,500 ppm were similar for the two standard waters at all concentration levels except 41,000 ppm (Figs. 5a and 5b, Welch's t-test, p>0.05), where the L2130i indicated differences in $\delta^2H$ deviation for the two different standard waters (Fig. 5b, Welch's t-test, p<0.05). This finding indicates that the L2130i's $\delta^2H$ accuracy for high moisture like 41,000 ppm is dependent on the isotopic composition, such as has been found for low moisture conditions below 4,000 ppm (Weng et al., 2020). This result further indicates that more than one standard water needs to be used in the field under not only low moisture but also high moisture conditions.

Additionally, the isotope dependence of $\delta^2$H accuracy may have related to the low suction flow rate of the L2130i (Thurnherr et al., 2020).

The [H$_2$O]-dependence of $\delta^{18}$O and $\delta^2$H accuracy also gives rise to uncertainty in deuterium excess (hereinafter called d-excess, d-excess = $\delta^2$H - 8$\delta^{18}$O) values, estimated with the uncorrected $\delta^{18}$O and $\delta^2$H values (Figs. 5c and 5f). The d-excess

deviation of the L2130i analyser significantly increased in a negative direction with concentration level on each standard water, and reached a maximum negative median value of -1.62 ‰ for DI1 and of -1.70 ‰ for DI2 (Fig. 5c). According to the calculation of d-excess, the decrease in d-excess with H$_2$O concentration mostly stemmed from the increase in the L2130i's $\delta^{18}$O values with H$_2$O concentration (Figs. 5a and 5c), which underlines the need for correcting for the [H$_2$O]-dependence, especially for $\delta^{18}$O accuracy at high concentration levels using the L2130i analyser.

The L1102i also had strong [H$_2$O]-dependence for both isotopes, larger than that of the L2130i (Figs. 5a-b and 5d-e). The larger variations also led to large deviations in the d-excess values (Fig. 5f). In addition, both $\delta^{18}$O and $\delta^2$H accuracy for the L1102i depend on isotopic compositions ($\delta^{18}$O: 36,000 ppm, $\delta^2$H: all concentration levels; Figs. 5d and 5e, Welch's t-test, p<0.05), different from the L2130i analyser. The above findings indicate that for the L1102i both $\delta^2$H and $\delta^{18}$O accuracy depend on H$_2$O concentration and the isotopic compositions, thus making the [H$_2$O]-dependence correction for both the $\delta^2$H

and $\delta^{18}$O accuracy using different standard waters necessary.

The L1102i's result of $\delta^{18}$O and $\delta^2$H deviations were comparable with those reported by Tremoy et al. (2011) who tested [H$_2$O]-dependence on $\delta^{18}$O and $\delta^2$H accuracy for L1102i up to 39,000 ppm against the reference H$_2$O concentration at 20,000 ppm. However, Tremoy et al. (2011) observed negative $\delta^{18}$O deviations at 39,000 ppm with a range between -2 and 0 ‰, different from this study. In addition, they showed a smaller increase in $\delta^2$H deviations with H$_2$O concentration from 20,000

to 39,000 ppm than this study. The above differences in $\delta^{18}$O and $\delta^2$H deviations between Tremoy et al. (2011) and this study shows that [H$_2$O]-dependence of $\delta^{18}$O and $\delta^2$H accuracy must be evaluated for each individual analyser (Aemisegger et al., 2012; Bailey et al., 2015).

In summary, the measurement accuracy for in $\delta^{18}$O and $\delta^2$H is more dependent on H$_2$O concentration for the L1102i than the L2130i, likely due to the older fitting algorithm for the initial version of the L1102i. In other words, the accuracy of [H$_2$O]-

dependence of $\delta^{18}$O and $\delta^2$H for the L2130i has been improved due to the updated/corrected fitting algorithm (Aemisegger et al., 2012), but our results still remind us of the importance of correcting for the [H$_2$O]-dependence of $\delta^{18}$O and $\delta^2$H accuracy for the L2130i analyser, particularly for high moisture condition at 36,000 ppm and above.

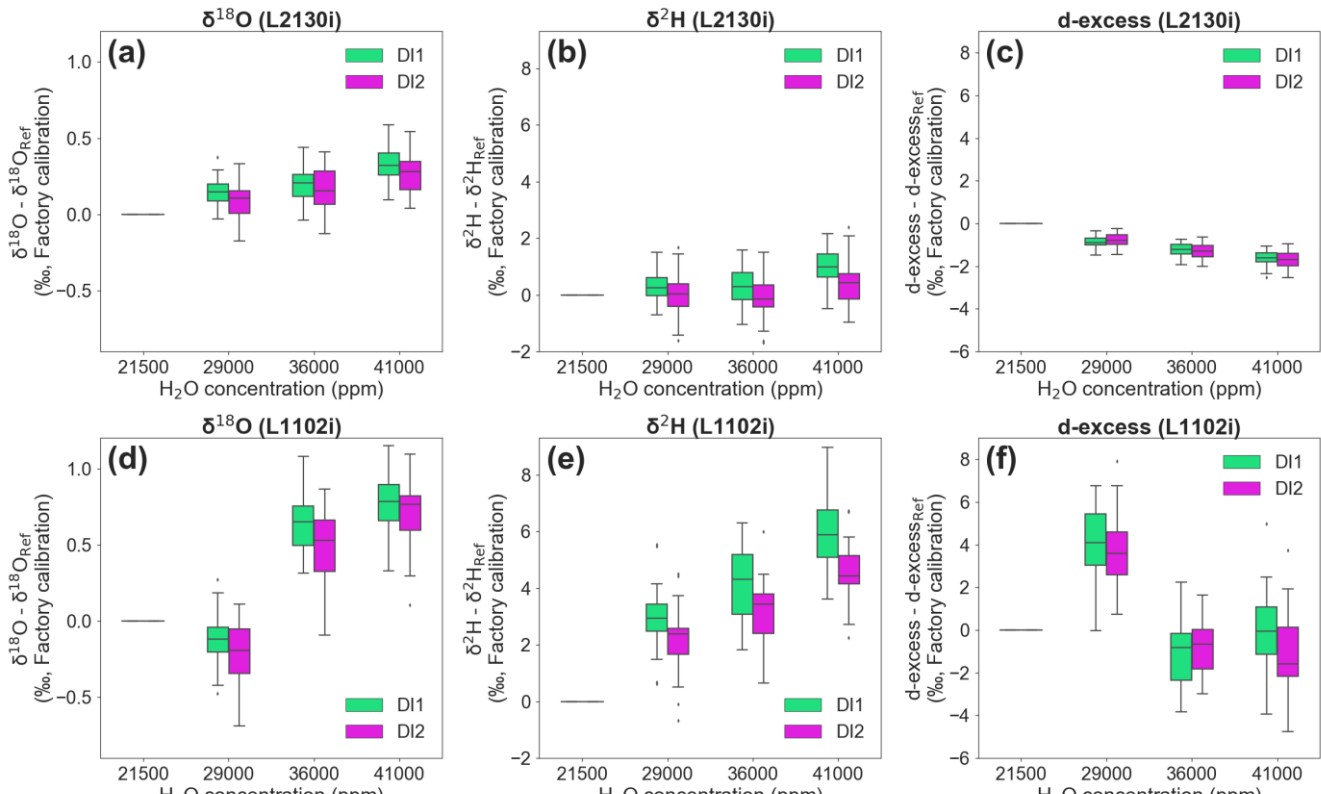

*Figure 5* Deviations of stable isotopic compositions ($\delta^{18}O$, $\delta^2H$ and d-excess) for each standard water (DI1 and DI2) at three different $H_2O$ concentrations compared to the 21,500 reference $H_2O$ concentration. Boxplots of (a) $\delta^{18}O$, (b) $\delta^2H$, and (c) d-excess deviations, measured by L2130i. Boxplots of (d) $\delta^{18}O$, (e) $\delta^2H$, and (f) d-excess deviations, measured by L1102i. The $\delta^{18}O$, $\delta^2H$ and d-excess values at 21,500 ppm are assigned a value of 0, and [$H_2O$]-dependence for each isotope can be observed as the deviation between the value at each concentration level from the value measured at 21,500 ppm.

### 3.3 Strategy for calibration of isotope values for water vapour concentration dependence

The four calibration strategies for correcting isotope values using measured $H_2O$ concentrations are compared in Figure 6. Across methods, the L2130i analyser usually displays lower median $\delta^{18}O$ RMSE values compared to the L1102i analyser (Fig. 6). This tendency is also found in $\delta^2H$ RMSE results on each calibration strategy (see Fig. S1 in the Supplement). The lower RMSE values of the L2130i analyser are mainly due to the higher precision of the L2130i analyser compared to the L1102i analyser.

The lowest median RMSE for the L1102i analyser is the cubic fitting method (Figs. 6b, 6d, 6f and 6h). Among all the calibration strategies of the L1102i analyser, the DI1-2*2Pairs calibration strategy with the cubic fitting method usually shows the minimum median RMSE value for $\delta^{18}O$ accuracy (Fig. 6h). For $\delta^2H$ accuracy of the L1102i analyser, the DI1-2*2Pairs calibration strategy also usually displays lower median RMSE values relative to the other strategies. These results indicate that this calibration strategy is most appropriate for correcting [$H_2O$]-dependence and improving the accuracy of isotope measurements with the L1102i analyser.

Compared with the L1102i analyser, the DI1-2*2Pairs calibration strategy of the L2130i analyser does not show clearly reduced isotopic RMSE values relative to the other calibration strategies, but still displays low RMSE values with a small distribution from each fitting method at each interval (Figs. 6a, 6c, 6e, 6g and S1). This indicates that the DI1-2*2Pairs strategy can be utilized for correcting [$H_2O$]-dependence of the L2130i analyser as well as the L1102i analyser. Since the calibration system uses the same tube line between the vaporizer and the branch point before the inlet port of each CRDS

analysers (Fig. 1a), we decided to utilize the DI1-2*2Pairs strategy for the L2130i analyser in the same way as the L1102i analyser.

Each CRDS analyser using the DI1-2*2Pairs strategy shows the lowest median RMSE values of isotopic accuracy at the shortest interval (28 h), which only slightly increases with interval over 8 days (Figs. 6g and 6h). This indicates that one [H$_2$O]-dependence calibration per week is enough to maintain good isotopic measurement of the CRDS analysers for an in-situ continuous observation remotely. According to the recent studies (Bonne et al., 2019; Weng et al., 2020), new CRDS models (e.g., L2130i and L2140i) did not show any significant changes for [H$_2$O]-dependence from ~500 to 25,000 ppm over several months up to 2 years. The consistency of [H$_2$O]-dependence may be extended to higher moisture conditions than 25,000 ppm, but based on our laboratory experiments, we decided to conduct the [H$_2$O]-dependence calibration at weekly or less interval with the DI1-2*2Pairs strategy. After we continuously run the ambient measurement and calibration systems at the ATTO site over several months, we will try to reduce the [H$_2$O]-dependence calibration frequency to maximize the ambient sampling period while maintaining high measurement accuracy of both CRDS analysers.

For both analysers using the DI1-2*2Pairs strategy, the RMSE distribution of isotopic accuracy gradually gets smaller with interval over 8 days (Figs. 6g and 6h). The smaller RMSE distributions at long intervals result from larger sample numbers for calculating RMSE values at long intervals, whereas the larger RMSE distributions at short intervals are attributed to smaller sample numbers for calculating RMSE (e.g., 28 h with ID-[3,4] & [7,8]: n=8 (4 samples × 2 measurement cycles, i.e., ID-5,6), 196 h with ID-[3,4] & [31,32]: n=104 (4 samples × 26 measurement cycles, i.e., ID-5-30); c.f., Fig. 2b).

The respective CRDS analyser with the DI1-2*2Pairs strategy also shows a similar range of RMSE among the five curve-fitting methods (Figs. 6g-h and S1g-h). The results do not clearly indicate which fitting method most frequently obtains the lowest RMSE values for each [H$_2$O]-dependence calibration interval. Hence, for the DI1-2*2Pairs strategy we obtained a [H$_2$O]-dependence fitting method, which only obtained minimum RMSE values of $\delta^{18}$O and $\delta^2$H, from each two calibration pairs at each interval and then calculated a contribution rate of the respective five [H$_2$O]-dependence fittings (i.e., linear, quadratic, cubic, quartic, linear surface fitting methods) to the total calibration pairs at each interval (Fig. 7). The linear-surface fitting method most frequently gives the lowest isotopic RMSE values, except $\delta^{18}$O accuracy of the L1102i analyser (i.e., cubic fitting method), at 28 h interval (Fig. 7). This indicates that the linear-surface fitting method at 28 h interval reduces uncertainties in correcting [H$_2$O]-dependence on isotopic accuracy most effectively, except for $\delta^{18}$O accuracy of the L1102i analyser. Compared with 28 h interval, the 2D fitting methods are the most appropriate for calibrating [H$_2$O]-dependence on isotopic accuracy over long intervals, excluding $\delta^2$H accuracy of the L1102i analyser (Fig. 7). The difference in fitting methods between 28 h and longer intervals suggests that the 28 h interval strategy can correct for both [H$_2$O]- and isotope-dependent errors by using the 3D fitting method, whereas the long interval strategies can correct for only [H$_2$O]-dependent errors with the 2D fitting methods.

Figure 8 presents the corrected isotope deviations by the best fitting methods (L2130i-$\delta^{18}$O: linear fitting, L2130i-$\delta^2$H: quadratic fitting, L1102i-$\delta^{18}$O: cubic fitting, L1102i-$\delta^2$H: linear-surface fitting) at weekly (i.e., 168 h) interval by assuming the continuous operation at the ATTO site, discussed above. The isotope deviations of each CRDS analysers do not substantially vary with H$_2$O concentration. This indicates the calibrations successfully corrected [H$_2$O]-dependence of isotope accuracy for each analysers. Based on Moreira et al. (1997), water vapour isotope values in Amazon rainforest is expected to change diurnally by up to 2 ‰ ($\delta^{18}$O) or 4-8 ‰ ($\delta^2$H) with H$_2$O concentration. The diel isotope variations are higher than the corrected deviation values of each CRDS analyser (Fig. 8). This supports that both the CRDS analysers will detect diel or probably seasonal/interannual variations in water vapour isotopes in Amazon rainforest.

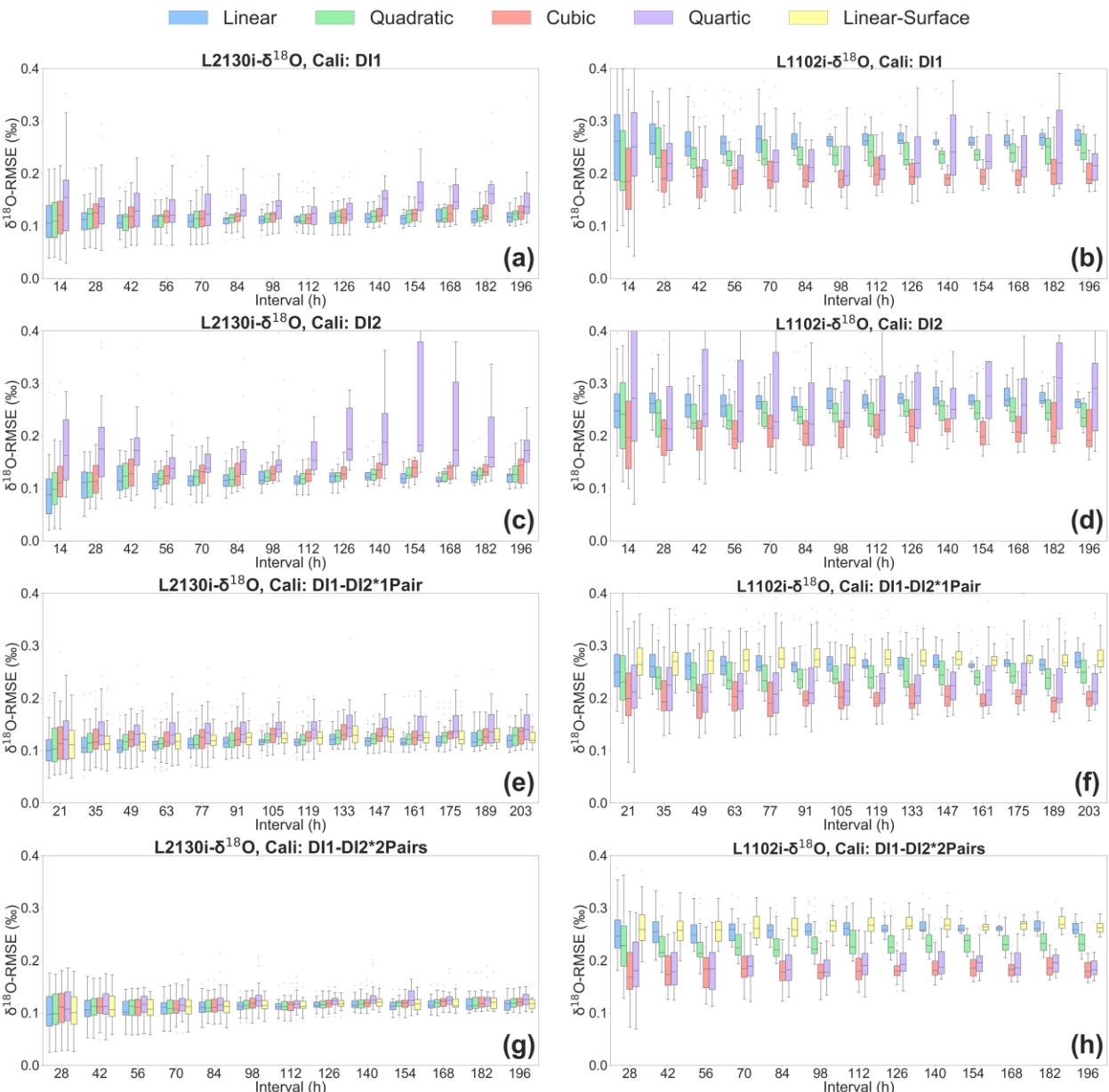

*Figure 6* Boxplots of root mean square error (RMSE) of $\delta^{18}$O, derived from calibrating [H$_2$O]-dependence of $\delta^{18}$O measurements by each of five fitting methods (i.e., linear, quadratic, cubic, quartic, linear surface fitting methods) for each of four calibration strategies: DI1, DI2, DI1-DI2*1Pair, DI1-2*2Pairs. Boxplots of (a) L2130's and (b) L1102's $\delta^{18}$O RMSE for the DI1 strategy, depending on interval length (i.e., the time period used for calibrating [H$_2$O]-dependence). Boxplots of (c) L2130's and (d) L1102's $\delta^{18}$O RMSE for the DI2 strategy, depending on interval length. Boxplots of (e) L2130's and (f) L1102's $\delta^{18}$O RMSE for the DI1-DI2*1Pair strategy, depending on interval length. Boxplots of (g) L2130's and (h) L1102's $\delta^{18}$O RMSE for the DI1-2*2Pairs strategy, depending on interval length. The procedure for assessing [H$_2$O]-dependence uncertainties (RMSE) is described in section 2.3.

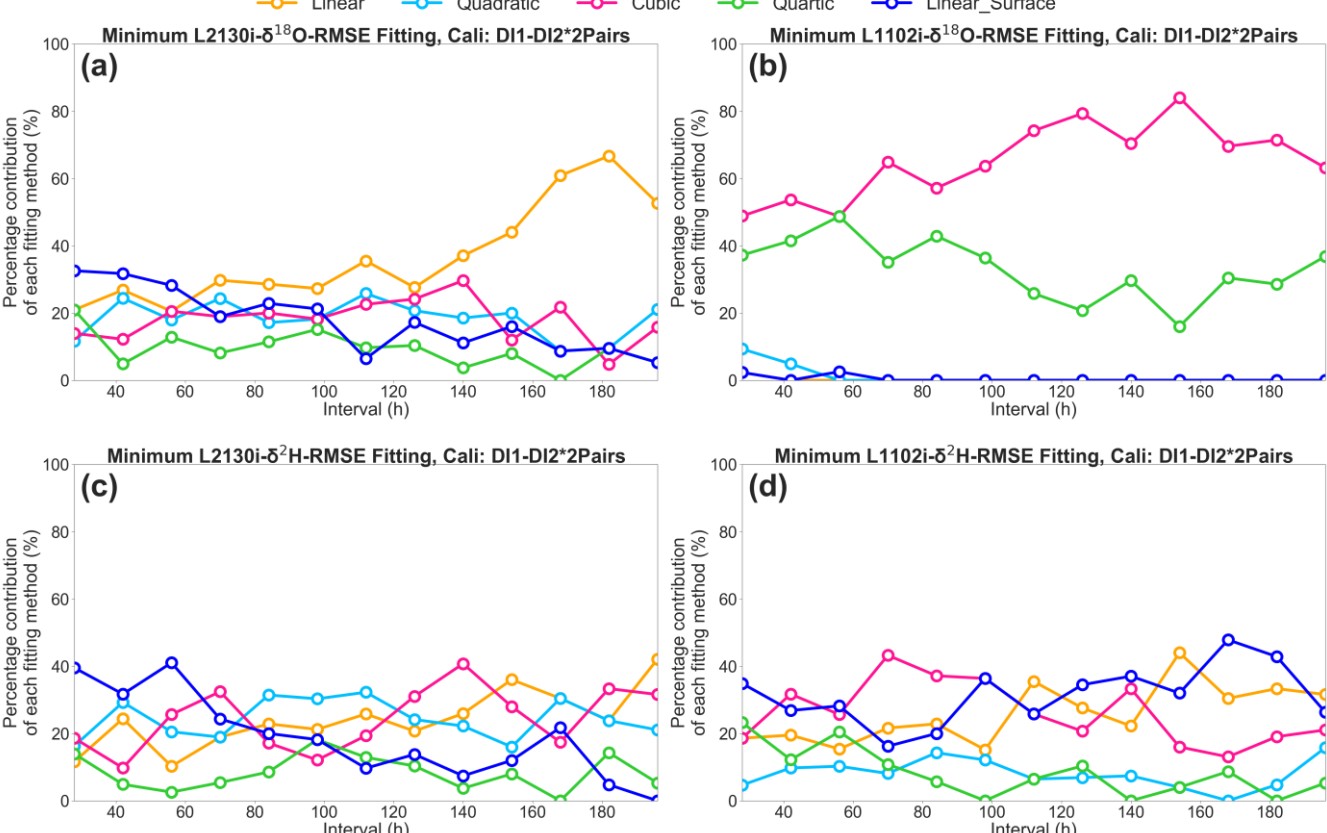

*Figure 7* Percentage contribution of the respective five [H$_2$O]-dependence fittings (i.e., linear, quadratic, cubic, quartic, linear surface fitting methods) to the total calibration pairs, which only obtained minimum RMSE values of $\delta^{18}$O and $\delta^2$H, at each interval for the DI1-2*2Pairs calibration strategy. (a) L2130i's and (b) L1102i's results for $\delta^{18}$O RMSE. (c) L2130i's and (d) L1102i's results for $\delta^2$H RMSE.

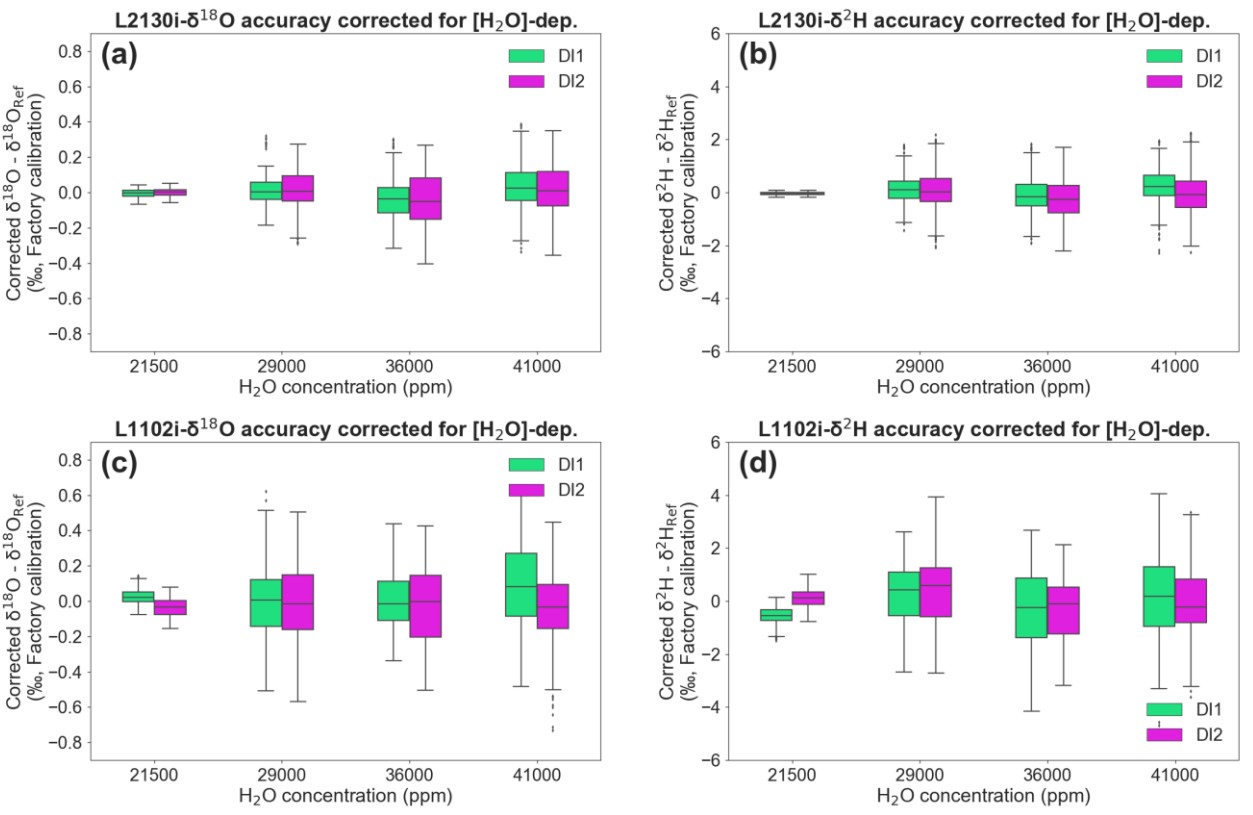

*Figure 8* Deviations of $\delta^{18}O$ and $\delta^2H$, both corrected for [H$_2$O]-dependence, for each standard water (DI1 and DI2) at four different H$_2$O concentrations. Boxplots of (a) $\delta^{18}O$ and (b) $\delta^2H$ deviations, corrected for [H$_2$O]-dependence of L2130i. Boxplots of (c) $\delta^{18}O$ and (d) $\delta^2H$ deviations, corrected for [H$_2$O]-dependence of L1102i. The [H$_2$O]-dependence of each CRDS analysers was corrected by the DI1-2*2Pairs calibration strategy with the best fitting methods (L2130i-$\delta^{18}O$: linear fitting, L2130i-$\delta^2H$: quadratic fitting, L1102i-$\delta^{18}O$: cubic fitting, L1102i-$\delta^2H$: linear-surface fitting) at 168 h interval.

## 4    Conclusions

This study extends previous work documenting water vapour concentration dependence of Picarro CRDS analysers to high moisture (> 35,000 ppm H$_2$O) likely to be measured in the Amazon rainforest and other tropical areas. We assessed the precision and accuracy of two CRDS analysers (i.e., model L1102i and L2130i) for concentration and isotopic measurements by using a custom-made calibration unit that regularly supplied standard water vapour samples at four different H$_2$O concentrations between 21,500 and 41,000 ppm to the CRDS analysers. Our results demonstrate that the newer version of the analyser (L2130i) has better precision for both $\delta^{18}O$ and $\delta^2H$ measurements under all H$_2$O concentration levels compared to the older model (L1102i). In addition, isotope measurements in both analysers varied with H$_2$O concentration, especially at H$_2$O concentration over 36,000 ppm. The concentration dependence of the L1102i analyser was stronger than the L2130i analyser. These findings indicate that calibrating the [H$_2$O]-dependence of $\delta^{18}O$ and $\delta^2H$ measurements for both the CRDS analysers during field deployment in high atmospheric moisture areas such as tropical forests is important.

Assuming continuous in situ observation together with regular calibration in tropical Amazon rainforest, we devised four calibration strategies, adjusted to our custom-made calibration system, and then evaluated which [H$_2$O]-dependence calibration procedure best improved the accuracy of $\delta^{18}O$ and $\delta^2H$ measurements for both the L2130i and L1102i analysers. The best [H$_2$O]-dependence strategy was the DI1-2*2Pairs strategy that required two pairs of a two-point calibration with different concentration levels from 21,500 to 41,000 ppm. The 28 h interval strategy with the linear-surface fitting method leads to the most accurate measurements for both the CRDS analysers, except $\delta^{18}O$ accuracy of the L1102i analyser that required the cubic fitting method. In addition, [H$_2$O]-dependence calibration uncertainties hardly changed at any interval over 8 days. That indicates one [H$_2$O]-dependence calibration per week is sufficient for correcting moisture-biased isotopic accuracy of the CRDS analysers. The best calibration strategy at weekly interval also supported that both CRDS analysers can sufficiently distinguish temporal variations of water vapour isotopes in the aimed ATTO site. In addition, since the recent studies indicate the consistency of the [H$_2$O]-dependence for the CRDS analysers over several months up to 2 years, we intend to determine the appropriate frequency for calibrating [H$_2$O]-dependence under high moisture conditions after a continuous operation over several months at the ATTO site to maximize the ambient sampling period.

## Data availability

All the data used in this publication are freely available at the "https://dx.doi.org/10.17617/3.4n."

## Author contribution

The calibration system was designed and developed by SK, JL, TS and US. The laboratory experiments were conducted by SK with assistance from JL, TS, US and FK. The IRMS analysis was done by HM and HG. DW helped us to check water vapour concentration in the ATTO site. SK conducted the data-analysis and wrote the manuscript with assistance from JL, FK, and HM. SK wrote the manuscript with contributions from all co-authors.

**Competing interests**

The authors declare that they have no conflict of interest.

**Acknowledgements**

This research was supported by German-Brazilian project ATTO, supported by the German Federal Ministry of Education
and Research (BMBF contracts 01LB1001A and FKR 01LK1602A) and the Brazilian Ministério da Ciência, Tecnologia e
Inovação (MCTI/FINEP contract 01.11.01248.00) as well as the Max-Planck Society. We are grateful to Susan Trumbore
(MPI-BGC) for reviewing the earlier manuscript and her valuable feedback, to Stefan Wolff and Matthias Sörgel (MPI-
Chemistry) for helping us to check $H_2O$ concentration in the ATTO site, and also to Jürgen M. Richter (MPI-BGC) for
helping us to prepare the DI1 and DI2 standard waters.

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
