# Peer review of "Characterising water vapour concentration dependence of commercial cavity ring-down spectrometers for continuous onsite atmospheric water vapour isotope measurements in the tropics"

_Atmospheric Measurement Techniques, 2020_

## Referee Comment (RC1) · Anonymous Referee #1 · 13 Nov 2020

Review of "Characterising water vapour concentration dependence of commercial cavity ring-down spectrometers for continuous onsite atmospheric water vapour isotope measurements in the tropics" Komiya et al. AMT

Cavity ring-down spectrometers have been increasingly used in water cycle dynamics studies that employ stable water isotopes as tracers for moist processes in the atmosphere and at the land-atmosphere interface. This paper presents an interesting evaluation of the water vapour mixing ratio dependency of cavity ring-down laser spectrometers focusing on the high specific humidity range that is encountered in tropical regions such as the Amazon.

In general, I found this paper interesting and mostly well written, with some instances, where it was more difficult to follow in particular in Section 2.3 about the calibration strategy of the water vapour mixing ratio dependence as well as the results part. I very much liked the precise description of the self-made calibration unit and find the approach of the authors very nice. Maybe they could add a bit more information on the long-term stability of their calibration system as mentioned in my minor comments below.

I recommend publication of this manuscript after the following three major comments and several minor comments below have been adequately addressed.

**Major comment:**

- The description of the calibration strategy in Section 2.3, especially with respect to Figures 2 and 3 was very difficult to follow. It would help if Figure 3 had panel numbers to which the captions could refer to. In general the figure captions could be improved to help the reader understand the Figures. Also the text should guide the reader better in understanding Figs. 2 and 3.
- 2) The results would benefit from a better structure within the sections. In particular, I would find it nice if the general findings valid for both instruments would be presented first and then the details about the L1102 and L2130.
- 3) The final recommendation of the paper to perform weekly or even more regular water vapour mixing ratio calibrations is surprising (e.g. P 16, L1-3), because an overwhelming majority of studies until now found that the water vapour mixing ratio dependencies of cavity ring-down systems from Picarro remain relatively constant in time and only occasional full water vapour mixing ratio dependency experiments were necessary, mainly for monitoring purposes. I therefore think the authors should discuss their recommendation and finding in this context of the existing literature. How much do you gain in permil uncertainty reduction for d18O and d2H (and maybe dexcess) with performing regular water vapour mixing ratio dependent calibrations compared to just applying a drift correction and a water vapour mixing ratio dependency correction that is constant in time? This is important because a lot of ambient measurement time is lost with the time consuming calibration scheme proposed by the authors.

**Minor comments:**

P 1, L 15: "including..." I would slightly rephrase to "which includes the correction of the H2O concentration dependence of isotope measurements"

P 1, L16: "past studies have assessed the [H2O]-dependence..."

P 1, L23: "... two pairs of a two point calibration with four different H2O concentration levels" sounds a bit obscure to me in an abstract. I think I see what you mean after having read the paper several times, but, maybe a more explicit formulation would be helpful P 1, L29: Overall very clear and nice abstract. I would find it nice to finish it with a less technical and more general sentence on a scientific level. This could be for example that this study shows that measurements in the tropics are in principle possible also at very high humidity levels, which has promising implications for water cycle studies focusing on tropical regions. But maybe the authors have a better idea for such a final zoom out sentence.

P 1, L 35: Maybe Dansgaard 1964 and Craig and Gordon 1965 would be good studies to cite here in addition to the more recent review by Galewsky et al. 2016.

P 1, L36: Here maybe specific modelling studies would be good to cite: Risi et al. 2010, Werner et al. 2011, Pfahl and Wernli 2012 instead of a review paper. I think the statement "has also improved simulations of hydrometeorological fields" could potentially be misunderstood. Isotopes are implemented in different global and regional circulation models to improve our understanding of how stable water isotopes are transported in the atmosphere and affected by phase changes in clouds, below the clouds and how they behave in different situation of surface-atmosphere interactions. Of course, the idea is to learn more about these moist processes through these isotope modelling studies. But, stable water isotopes are implemented as *passive* tracers in these models and do not directly impact other hydrometeorological fields.

P 2, L6: Maybe here the polar regions, midlatitues and subtropics/tropics could be mentioned separately. There are many studies based on water vapour isotope measurements available in polar and midlatitude regions, some in the subtropics (e.g. Gonzalez et al. 2016, Bailey et al. 2013) but only very few in the tropics (e.g. Tremoy et al. 2012, Aemisegger et al. 2020). In particular studies over tropical continental regions are rare.

P 2, L16: Replace "confirmed" by "showed" because this was not shown before, and remove s in "an older model".

P 2, L 26: This statement about the diel cycle comes a bit abruptly after the discussion about older studies of water vapour isotopes in different humidity ranges. Maybe a new paragraph would help and a smoother transition from mainly instrument technical aspects to previous observations in nature.

P 2, L 33: ATTO is a new abbreviation to me

P 2, L35: Old and new instead of older and newer, the latter would require a "with respect to what" statement

P 2, L42: "sufficiently detect..." sounds a bit awkward. Maybe something like "based on the uncertainty quantification presented, we discuss whether the CRDS analysers can detect natural signals of ...".

P 3, L16: To which water vapour mixing ratio level does this dew point correspond?P 3, Section 2.1: In general: very nice self-made setup and description of it! I would be curious to see it in operation. A picture of the full setup next to the schematic in Fig. 1 would

be great! Also could the authors comment on the long-term stability of their system? Are there any failure-prone parts in this system? E.g. how about the Syrringe pump? P 3, L 26: Could you indicate the total flow rate of the CRDS instruments? A recent study has shown that the water vapour mixing ratio dependency is actually depending on the instrument's flow rate (see supplement of Thurnherr et al. 2020)

P 3, L35: Could the authors shortly mention how this correction is implemented? Were the two CRDS instruments also connected to a dew point generator at different dew point temperatures to ensure that no bias affects their water vapour concentration?

P 4, L13: Also use humidity or water vapour mixing ratio level and not "moisture level" P7: I had difficulties to understand Fig. 2. Is this actually a Figure or a Table? Maybe the caption could be a bit more detailed. And why does the DI1 strategy contain Cali. ID 1 to 3 and not only 1 and 3? Same for DI2, why is it not 2 & 4? And why do I see Cali IDs such as 43-47 at the 14h calibration interval. So in short, I think, I get the general idea, but I was confused by some details. The text and Figure/Table could also speak a bit more together to help the reader here.

P 9, L 20: At these water vapour mixing ratio levels, the instrument is measuring at its uppermost limits. So, I am not sure if the larger variability in H2O concentrations should be attributed to the calibration unit or a saturation effect within the cavity. Are the instruments used here optimized factory-wise (e.g. absorption peak scanning strategy) for high water vapour mixing ratio ranges? If yes, then it would be good to mention this in the methods section.

P 9, L24-25: I was confused here: variability in H2O concentration higher than what? P 9, L27: The higher  $\delta^{18}$ O and  $\delta^{2}$ H precision at higher H2O levels is a bit surprising. I would have expected a saturation effect at some point.

P 9, L34: This is what I would have expected, maybe a slight rearrangement of paragraphs would be good here first discussion statements that are valid for both instruments and then discussing the individual instrument versions.

P 9, L37: I would make it clear that you mean the absorption peak fitting algorithm and you could add the "absorption peak scanning strategy".

P 11, L 9: Replace "checked" by "tested"

P 11, L 12: Replace "confirmed" by "showed"

P 12, L 6: Aren't the lower RMSE values for the L2130 compared to the L1102 mainly due to the higher precision of the L2130?

P 15, L 9: And in other tropical areas to make it more general?

P 15, L13: Also here I would rather use "water vapour mixing ratio level" instead of moisture (moisture is more general and could also for instance imply liquid water).

**References**

Aemisegger, F., Vogel, R., Graf, P., Dahinden, F., Villiger, L., Jansen, F., Bony, S., Stevens, B., and Wernli, H.: How Rossby wave breaking modulates the water cycle in the North Atlantic trade wind region, Weather Clim. Dynam. Discuss., https://doi.org/10.5194/wcd-2020-51, in review, 2020.

Craig, H. and Gordon, L. I.: Deuterium and oxygen 18 variations in the ocean and the marine atmosphere, 1965.

Bailey, A., Toohey, D., and Noone, D.: Characterizing moisture exchange between the Hawaiian convective boundary layer and free troposphere using stable isotopes in water, J. Geophys. Res. Atmos., 118, 8208–8221, doi:10.1002/jgrd.50639, 2013

Dansgaard, W.: Stable isotopes in precipitation. Tellus, 16: 436-468. https://doi.org/10.1111/j.2153-3490.1964.tb00181.x, 1964.

Galewsky, J., Steen-Larsen, H. C., Field, R. D., Worden, J., Risi, C., and Schneider, M.: Stable isotopes in atmospheric water vapor and applications to the hydrologic cycle, Rev. Geophys., 54, 809–865, doi:10.1002/2015RG000512, 2016.

González, Y., Schneider, M., Dyroff, C., Rodríguez, S., Christner, E., García, O. E., Cuevas, E., Bustos, J. J., Ramos, R., Guirado-Fuentes, C., Barthlott, S., Wiegele, A., and Sepúlveda, E.: Detecting moisture transport pathways to the subtropical North Atlantic free troposphere using paired H2O-δD in situ measurements, Atmos. Chem. Phys., 16, 4251–4269, https://doi.org/10.5194/acp-16-4251-2016, 2016.

Pfahl, S., Wernli, H., and Yoshimura, K.: The isotopic composition of precipitation from a winter storm – a case study with the limited-area model COSMOiso, Atmos. Chem. Phys., 12, 1629–1648, https://doi.org/10.5194/acp-12-1629-2012, 2012.

Risi, C., Bony, S., Vimeux, F., and Jouzel, J.: Water stable isotopes in the LMDZ4 general circulation model: Model evaluation for present day and past climates and applications to climatic interpretation of tropical isotopic records, J. Geophys. Res., 115, D12118, doi:10.1029/2009JD013255, 2010.

Tremoy, G., Vimeux, F., Mayaki, S., Souley, I., Cattani, O., Risi, C., Favreau, G., and Oi, M.: A 1-year longδ18O record of water vapor in Niamey (Niger) reveals insightful atmospheric processes at different timescales, Geophys. Res. Lett., 39, L08805, doi:10.1029/2012GL051298, 2012.

Thurnherr, I., Kozachek, A., Graf, P., Weng, Y., Bolshiyanov, D., Landwehr, S., Pfahl, S., Schmale, J., Sodemann, H., Steen-Larsen, H. C., Toffoli, A., Wernli, H., and Aemisegger, F.: Meridional and vertical variations of the water vapour isotopic composition in the marine boundary layer over the Atlantic and Southern Ocean, Atmos. Chem. Phys., 20, 5811–5835, https://doi.org/10.5194/acp-20-5811-2020, 2020.

Werner, M., Langebroek, P. M., Carlsen, T., Herold, M., and Lohmann, G.: Stable water isotopes in the ECHAM5 general circulation model: Toward high-resolution isotope modeling on a global scale, J. Geophys. Res., 116, D15109, doi:10.1029/2011JD015681, 2011.

---

## Referee Comment (RC2) · Anonymous Referee #2 · 18 Nov 2020

**Review of 'Characterising water vapour concentration dependence of commercial cavity ring-down spectrometers for continuous onsite atmospheric water vapour isotope measurements in the tropics' by Komiya *et al.*, submitted to AMT**

This manuscript presents a characterisation of concentration dependence of 2 commercial cavity ring-down spectrometers at 4 high water concentration levels between 21500 and 41000 ppm. The authors assessed the calibration results using 4 different calibration strategies, and for each calibration strategy using 5 different fitting functions. The authors concluded with identifying the most appropriate calibration strategy and fitting choice for their two specific instruments.

The relative new aspects of the study are the focus of the calibration at high water concentration levels (21500 to 41000 ppm) and the assessment of different calibration strategies (including fitting choices). The manuscript is relatively well written. Below I have detailed 2 major points, and some minor issues with suggestions.

**Major comments**

1. One key aspect of the study is to test the 5 calibration strategies. A general question would be: what motivates the authors to choose *these specific* calibration strategies in the first place? In other words, what is the purpose or hypothesis, advantage, and disadvantage behind each specific strategy? It would be more clear and helpful if the authors could briefly clarify these questions before presenting the methods and results.

2. The authors have described a custom-built calibration unit. The unit's working principle is similar to the commercially available (since 2013?) Picarro Standards Delivery Module (A0101). What is the specific motivation for the authors to build their own calibration unit? What main differences or advantages do the authors achieve?

**Detailed comments**

Page 1, Line 12: 'isotope ratios' is confusing here since it normally refers to R and not delta-notation. It may be replaced with 'isotope compositions'. The term 'isotope ratios' has been used in many places in the manuscript, and all should be reconsidered or replaced.

P 2, L 17: 'models' replaced to 'model '

P 2, L 26: To make the structure more clear, suggest to add 'In addition, ' before 'Moreira et al.'.

P 2, L 37: To make the sentence more clear, suggest to remove 'expected based on past measurement at the ATTO site' or place it in a new sentence.

P 3, L25: Maybe remove '(Fig. 1)'. This is already referred to in the beginning.

P 3, L40: Is the purpose to remove 'salt compounds'? I would suppose that there is very low salt contents in the fresh water (if the original water is fresh water and not sea water, otherwise it should be stated). I think the purpose is to in general remove contaminants.

P4, L21: The statement is very strong here so that it requires proof. If it is/can not be proved, it is rather a hypothesis or speculation. Also, this statement is in contradicted to the statement in P9, L15. Maybe replace 'any' to 'to a large extent the' or something like this.

P4, L26: Not really common in the literature to use "[H2O]" as abbreviation. "[H2O]" has appeared in many places to represent 'water concentration level'. I personally feel it hinders the reading flow. Maybe it is better to use 'WCL' or simply 'concentration'?

P4, L29: 'Thus, the isotopic deviation values at 21,500 ppm are set to 0.' Personally think this sentence is redundant and can be removed with no harm.

P4, L29: '(Fig. 2)' can be removed here.

P5, L7: '(Fig. 3)'. It is better to assign each subplot with a letter, and refer the description to the exact subfigure, for example '(Fig. 3a)'. With this context, suggest to add letters for all the subplots in Fig 3, 4, 6, and 7, as you have done for other figures.

P5, L16: 'For example,'

P5, L16: Suggest to remove '=', also for the other occasions throughout the manuscript.

P5, L26: Suggest to remove 'For instance,'.

P5, L35-38: The sentence is redundant. Suggest to change to 'For each calibration strategy, we also calculated RMSE value for each of the [H 2 O]-dependence 2D or 3D fittings'.

P6, Table 1: Remove the bold title.

P9, L12: '(1.1%).' to '(1.1%)'.

P9, L12: Suggest to remove '(> 40,000 ppm)'.

P9, L125: Should be 'lower' precision? I can see the standard deviation becomes larger at high concentrations.

P9, L126: Following above, should it be 'decrease'?

P9, L133: Remove 'At all H 2 O concentration levels, and '.

P9, L137: Again, the statement here sounds strong. It seems that it has been proved, but actually not in the manuscript. Therefore, I would say it is more like a speculation or hypothesis. Suggest to change 'mainly be' to 'is likely' or something similar.

Same applies to P11, L17.

P10, L8: Add space before unit '%' and check all other units to be consistent.

P10, L16: Should be 'Fig. 5a-d'.

P10, L17: Should be 'Fig. 5b'.

P10, L18-19: This is an interesting observation. There is recently also a systematic study on concentration dependence that emphasizes this isotope-dependency. Probably this is something worthy to be discussed?

Weng, Y., Touzeau, A., and Sodemann, H.: Correcting the impact of the isotope composition on the mixing ratio dependency of water vapour isotope measurements with cavity ring-down spectrometers, Atmos. Meas. Tech., 13, 3167–3190, https://doi.org/10.5194/amt-13-3167-2020, 2020.

P12, L10: Refer exactly to subfigures such as (Fig 6e, j).

P12, L23: It would be helpful to explain the method before instead of going directly to the results. It is not clear for the readers what have been calculated here and how they are calculated.

P12, L39: 'smaller sample numbers for calculating RMSE (28 h: n=43, 196 h: n=19; c.f., Fig. 2)'. It is confusing here because it seems to me the short-interval calibration has larger sample numbers (n=43).

P12, L31-39: This paragraph jumps back to Fig. 6. Suggest to move it up after L19, before presenting Fig 7.

P16, L2: Replace 'the CRDS' to 'CRDS'.

Fig S1: delta2H instead of delta18O in first line.

---

## Author Comment (AC3) · 4 Jan 2021

**Response to Reviewer #1**

Please note: the original reviewer's comments are printed in black, while our response is printed in orange.

Review of "Characterising water vapour concentration dependence of commercial cavity ring-down spectrometers for continuous onsite atmospheric water vapour isotope measurements in the tropics" Komiya et al. AMT

Cavity ring-down spectrometers have been increasingly used in water cycle dynamics studies that employ stable water isotopes as tracers for moist processes in the atmosphere and at the land-atmosphere interface. This paper presents an interesting evaluation of the water vapour mixing ratio dependency of cavity ring-down laser spectrometers focusing on the high specific humidity range that is encountered in tropical regions such as the Amazon.

In general, I found this paper interesting and mostly well written, with some instances, where it was more difficult to follow in particular in Section 2.3 about the calibration strategy of the water vapour mixing ratio dependence as well as the results part. I very much liked the precise description of the self-made calibration unit and find the approach of the authors very nice. Maybe they could add a bit more information on the long-term stability of their calibration system as mentioned in my minor comments below.

I recommend publication of this manuscript after the following three major comments and several minor comments below have been adequately addressed.

**Response**: We sincerely appreciate your support and comments for our study. We have revised our manuscript following your comments and suggestions. The revised parts of the manuscript are shown with track changes in the marked-up version. We also revised all the Figures.

**Major comment:**

1) The description of the calibration strategy in Section 2.3, especially with respect to Figures 2 and 3 was very difficult to follow. It would help if Figure 3 had panel numbers to which the captions could refer to. In general the figure captions could be improved to help the reader understand the Figures. Also the text should guide the reader better in understanding Figs. 2 and 3.

**Response**: We revised the Figs. 2 and 3, and also the captions. Following the revised Figs. 2 and 3, we revised section 2.3 (see pp. 8-9, Figure 2; pp. 10-11, Figure 3; pp. 5-6, section 2.3 in the marked manuscript).

2) The results would benefit from a better structure within the sections. In particular, I would find it nice if the general findings valid for both instruments would be presented first and then the details about the L1102 and L2130.

**Response**: Following your suggestions here and in the Minor comments section, we revised the paragraphs in section 3.1 (see pp. 11-12 in the marked manuscript). In addition, we rearranged the paragraphs in section 3.3 (see pp. 14-16 in the marked manuscript), also suggested by the RC2 reviewer's minor comment. Regarding section 3.2, we decided not to rearrange the paragraphs because in our opinion this would impact negatively the narrative flow.

3) The final recommendation of the paper to perform weekly or even more regular water vapour mixing ratio calibrations is surprising (e.g. P 16, L1-3), because an overwhelming majority of studies until now found that the water vapour mixing ratio dependencies of cavity ring-down systems from Picarro remain relatively constant in

time and only occasional full water vapour mixing ratio dependency experiments were necessary, mainly for monitoring purposes. I therefore think the authors should discuss their recommendation and finding in this context of the existing literature. How much do you gain in permil uncertainty reduction for d18O and d2H (and maybe dexcess) with performing regular water vapour mixing ratio dependent calibrations compared to just applying a drift correction and a water vapour mixing ratio dependency correction that is constant in time? This is important because a lot of ambient measurement time is lost with the time consuming calibration scheme proposed by the authors.

**Response**: Thank you for your suggestion, with which we agree. Our final recommendation in the original manuscript "to conduct the [$H_2O$]-dependence calibration **at 28 h or less interval**." (see p. 16, LL. 1-3 in the original manuscript) may indeed be too conservative and would not improve significantly the quality of the measurement, while taking away precious measurement time. As you correctly point out, recent studies show that "new CRDS models (e.g., L2130i and L2140i) did not show any significant changes for [$H_2O$]-dependence from ~500 to 25,000 ppm over several months up to 2 years." (see p. 15, LL. 11-13 in the marked manuscript). However, there was to our knowledge no study prior to ours that assessed the [$H_2O$]-dependence consistency above 25,000 ppm $H_2O$ over several months. According to the results of our laboratory experiments, we decided that "one [$H_2O$]-dependence calibration per week is sufficient for correcting moisture-biased isotopic accuracy of the CRDS analysers." (see p. 19, L. 1 in the marked manuscript). Therefore, we discussed this part in section 3.3 in Results and Discussion as follows: "According to the recent studies (Bonne et al., 2019; Weng et al., 2020), new CRDS models (e.g., L2130i and L2140i) did not show any significant changes for [$H_2O$]-dependence from ~500 to 25,000 ppm over several months up to 2 years. The consistency of [$H_2O$]-dependence may be extended to higher concentrations than 25,000 ppm, but based on our laboratory experiments, we decided to conduct the [$H_2O$]-dependence calibration at weekly or less interval with the DI1-2*2Pairs strategy. After we continuously run the ambient measurement and calibration systems at the ATTO site over several months, we will try to reduce the [$H_2O$]-dependence calibration frequency to maximize the ambient sampling period while maintaining high measurement accuracy of both CRDS analysers." (see p. 15, LL. 11-17 in the previous manuscript). Base on the revised discussion, we also revised the last sentences in section 4: Conclusion (see pp. 18-19 in the previous manuscript).

**Minor comments:**

P 1, L 15: "including…" I would slightly rephrase to "which includes the correction of the H2O concentration dependence of isotope measurements"

**Response**: We revised the text according to your suggestion (see p. 1, L. 15 in the marked manuscript).

P 1, L16: "past studies have assessed **the** [H2O]-dependence…"

**Response**: We revised the text according to your suggestion (see p. 1, L. 16 in the marked manuscript).

P 1, L23: "… two pairs of a two point calibration with four different H2O concentration levels" sounds a bit obscure to me in an abstract. I think I see what you mean after having read the paper several times, but, maybe a more explicit formulation would be helpful

**Response**: We revised this sentence as follows: "The best strategy required conducting a two-point calibration with four different $H_2O$ concentration levels, carried out at the beginning and end of the calibration interval." (see p. 1, LL. 24-25 in the marked manuscript).

P 1, L29: Overall very clear and nice abstract. I would find it nice to finish it with a less technical and more general sentence on a scientific level. This could be for example that this study shows that measurements in the tropics are in principle possible also at very high humidity levels, which has promising implications for water cycle studies focusing on tropical regions. But maybe the authors have a better idea for such a final zoom out sentence.

**Response**: Agreed. We added a general sentence on a scientific level as follows; "This study shows that the CRDS analysers, appropriately calibrated for [$H_2O$]-dependence, allow the detection of natural signals of stable water vapour isotopes at very high humidity levels, which has promising implications for water cycle studies in areas like the central Amazon rainforest and other tropical regions." (see pp. 1, LL. 30-32 in the marked manuscript).

P 1, L 35: Maybe Dansgaard 1964 and Craig and Gordon 1965 would be good studies to cite here in addition to the more recent review by Galewsky et al. 2016.

**Response**: Agreed - we added the suggested two references (see p. 1, L. 38 in the marked manuscript).

P 1, L36: Here maybe specific modelling studies would be good to cite: Risi et al. 2010, Werner et al. 2011, Pfahl and Wernli 2012 instead of a review paper. I think the statement "has also improved simulations of hydrometeorological fields" could potentially be misunderstood. Isotopes are implemented in different global and regional circulation models to improve our understanding of how stable water isotopes are transported in the atmosphere and affected by phase changes in clouds, below the clouds and how they behave in different situation of surface-atmosphere interactions. Of course, the idea is to learn more about these moist processes through these isotope modelling studies. But, stable water isotopes are implemented as *passive* tracers in these models and do not directly impact other hydrometeorological fields.

**Response**: Following you suggestion, we used the suggested three references, and also revised this sentence as follows: "Incorporating water vapour isotopic information into global and regional circulation models has also improved our understanding of how stable water isotopes are transported in the atmosphere and affected by phase changes in- and below-clouds, and how they behave in different situation of surface-atmosphere interactions (Risi et al., 2010; Werner et al., 2011; Pfahl et al., 2012)." (see p. 1, L. 39 – p. 2, L. 3 in the marked manuscript).

Based on this revision, in the next sentence we also corrected "hydrometeorological cycles" to "the atmospheric hydrological system" (see p. 2, L. 5 in the marked manuscript).

P 2, L6: Maybe here the polar regions, midlatitues and subtropics/tropics could be mentioned separately. There are many studies based on water vapour isotope measurements available in polar and midlatitude regions, some in the subtropics (e.g. Gonzalez et al. 2016, Bailey et al. 2013) but only very few in the tropics (e.g. Tremoy et al. 2012, Aemisegger et al. 2020). In particular studies over tropical continental regions are rare.

**Response**: According to your suggestion, we revised this sentence as follows: "So far, there are many studies based on field water isotopic measurements available in polar and midlatitude regions, some in the subtropics (e.g., Bailey et al., 2013; Gonzalez et al., 2016) but only very few in the tropics (e.g., Tremoy et al., 2012; Aemisegger et al., 2020). Particularly studies in tropical continental regions, such as the Amazon basin region, are rare." (see p. 2, LL. 11-15 in the marked manuscript).

P 2, L16: Replace "confirmed" by "showed" because this was not shown before, and remove s in "an older model".

**Response**: We replaced it with 'demonstrated' because this revision more fits here. (see p. 2, L. 25 in the marked manuscript).

P 2, L 26: This statement about the diel cycle comes a bit abruptly after the discussion about older studies of water vapour isotopes in different humidity ranges. Maybe a new paragraph would help and a smoother transition from mainly instrument technical aspects to previous observations in nature.

**Response**: Based on your and RC2 reviewer suggestions, we added 'in addition' before this sentence (see p. 2, L. 37 in the marked manuscript). Additionally, we started a new paragraph from "However, $H_2O$ concentrations within the Amazon…" based on the context (see p. 2, LL. 35-42 in the marked manuscript).

P 2, L 33: ATTO is a new abbreviation to me

**Response**: The abbreviation is mentioned for the first time and explained at p. 2 L. 23 of the original manuscript (and LL. 35-36 on p.2 in the marked version of the revised manuscript).

P 2, L35: Old and new instead of older and newer, the latter would require a "with respect to what" statement

**Response**: According to your suggestion, we revised as follows: "an old (L1102i) and a new CRDS models (L2130i)" (see p. 3, L.4 in the marked manuscript).

In addition, regarding "the latter would require a "with respect to what" statement", we added "both" before "an old (L1102i) and a new CRDS models (L2130i)" to clearly state it (see p. 3, L. 4 in the marked manuscript).

P 2, L42: "sufficiently detect…" sounds a bit awkward. Maybe something like "based on the uncertainty quantification presented, we discuss whether the CRDS analysers can detect natural signals of …".

**Response**: According to your suggestion, we revised the sentence as follows: "Based on the uncertainty quantification presented, we discussed whether the CRDS analysers with our calibration setup can sufficiently detect natural signals of stable water vapour isotopes expected at the ATTO site." (see p. 3, LL. 11-13 in the marked manuscript).

P 3, L16: To which water vapour mixing ratio level does this dew point correspond?

**Response**: We revised here as follows: "dew point temperature of -32 °C or below (~300 ppm of $H_2O$ or below)" (see p. 3, L. 37 in the marked manuscript).

P 3, Section 2.1: In general: very nice self-made setup and description of it! I would be curious to see it in operation. A picture of the full setup next to the schematic in Fig. 1 would be great! Also could the authors comment on the long-term stability of their system? Are there any failure-prone parts in this system? E.g. how about the Syrringe pump?

**Response**: Thank you for your support. We added two photos showing the main part of the dry air and calibration units (see p. 7, Figs. 1b and 1c in the marked manuscript). Also following the detailed information about the 3-way solenoid valves in Fig. 1c., we revised the solenoid valves' information in Fig. 1a, concisely presented in the previous manuscript. In addition, regarding the long-term stability, we added maintenance information for the syringe-pump and membrane dryer (see p. 3, LL. 26-28 and LL. 33-35 in the marked manuscript).

P 3, L 26: Could you indicate the total flow rate of the CRDS instruments? A recent study has shown that the water vapour mixing ratio dependency is actually depending on the instrument's flow rate (see supplement of Thurnherr et al. 2020)

**Response**: Yes, we added the total flow rate of the two CRDS analysers as follows: "… the suction flow rates of the two CRDS analyzers (~50 mL min$^{-1}$ in total)." (see p. 4, L. 8 in the marked manuscript). In addition, Thurnherr et al. (2020) found that L2130i's isotope dependence with humidity dependence occurred by not the high suction flow rate (300 mL min$^{-1}$) but the low suction flow rate (50 mL min$^{-1}$) like our study. So we cited this finding to explain why our L2130i analyzer detected the isotope dependence of $\delta^2$H accuracy in section 3.2 in Results and Discussions as follows: "Additionally, the isotope dependence of $\delta^2$H accuracy may have related to the low suction flow rate of the L2130i (Thurnherr et al., 2020)." )." (see p. 13, LL. 20-21 in the marked manuscript).

P 3, L35: Could the authors shortly mention how this correction is implemented? Were the two CRDS instruments also connected to a dew point generator at different dew point temperatures to ensure that no bias affects their water vapour concentration?

**Response**: Unfortunately we were not able to connect a dew point generator to the CRDS analysers during the experiment period. So, we used a nonlinear calibration fitting, determined by Winderlich et al., (2010) and recommended for old CRDS models (e.g., G1301 $CO_2$/$CH_4$/$H_2O$ analyser) of $H_2O$ measurement by Rella (2010), in order to correct for $H_2O$ concentration by the old CRDS model (e.g., L1102i). Therefore, we revised this sentence as follows:

"Since $H_2O$ concentration values measured by old CRDS models (e.g., L1102i) are biased due to the self-broadening effect of water vapour (Winderlich et al., 2010), $H_2O$ concentration measuring by the L1102i were corrected by a nonlinear calibration fitting, determined by Winderlich et al., (2010) and recommended for old CRDS models (e.g., G1301 $CO_2$/$CH_4$/$H_2O$ analyser) by Rella (2010)."(see p. 4, LL. 15-18 in the marked manuscript).

P 4, L13: Also use humidity or water vapour mixing ratio level and not "moisture level"

**Response**: We replaced "moisture level" with "concentration level" (see p. 4, L. 35 in the marked manuscript), and revised the corresponding parts throughout the manuscript.

P7: I had difficulties to understand Fig. 2. Is this actually a Figure or a Table? Maybe the caption could be a bit more detailed. And why does the DI1 strategy contain Cali. ID 1 to 3 and not only 1 and 3? Same for DI2, why is it not 2 & 4? And why do I see Cali IDs such as 43-47 at the 14h calibration interval. So in short, I think, I get the general idea, but I was confused by some details. The text and Figure/Table could also speak a bit more together to help the reader here.

**Response**: Since we combined the top diagram with a table at the bottom, we assigned those as a Figure. To make it clearer, we added a figure letter to each part (see p. 8, Figs. 2a and 2b in the marked manuscript).

We are sorry that the Cali. ID information in the DI1, DI2 and DI1-DI2*1Pair strategies in the bottom table was misleading to readers, so we revised the corresponding parts (see p. 8, Fig. 2b in the marked manuscript).

P 9, L 20: At these water vapour mixing ratio levels, the instrument is measuring at its uppermost limits. So, I am not sure if the larger variability in H2O concentrations should be attributed to the calibration unit or a saturation effect within the cavity. Are the instruments used here optimized factory-wise (e.g. absorption peak scanning strategy) for high water vapour mixing ratio ranges? If yes, then it would be good to mention this in the methods section.

**Response**: Agreed. We revised this sentence as follows: "In addition, the larger variation in $H_2O$ concentration at 41,000 ppm may have been influenced by instability of the calibration system and a saturation effect inside the cavity measuring $H_2O$ concentration near the upper limit (50,000 ppm)." (see p. 11, LL. 22-24 in the marked manuscript).

Our CRDS analysers just have a factory standard setting for absorption peak scanning strategy, so we did not mention it in the Material and Methods.

P 9, L24-25: I was confused here: variability in H2O concentration higher than what?

**Response**: We revised here as follows: "…even though variability in $H_2O$ concentration measurement was higher at 29,000 ppm ($H_2O$-$\sigma \geq$ 578.6 ppm) than at 21,500 ppm ($H_2O$-$\sigma \leq$ 253.5 ppm)." (see p. 11, LL. 36-37 in the marked manuscript).

P 9, L27: The higher d$^{18}$O and d$^2$H precision at higher H2O levels is a bit surprising. I would have expected a saturation effect at some point.

**Response**: Indeed, we also did not expect it at first, yet this is what we have observed in our experiment (see p. 11, L. 34 – p. 12, L. 2 in the marked manuscript). Future work may provide an explanation.

P 9, L34: This is what I would have expected, maybe a slight rearrangement of paragraphs would be good here first discussion statements that are valid for both instruments and then discussing the individual instrument versions.

**Response**: Following your suggestion, we rearranged the second and last paragraphs in section 3.1 (see p. 11, L. 27 – p. 12, L. 2 in the marked manuscript). We did not rearrange the first paragraph because the first paragraph does not describe anything about Table 1. In the second paragraph, based on Table 1's results we described the general findings for both analysers, and then discussed more detailed parts in the third paragraph.

P 9, L37: I would make it clear that you mean the absorption peak fitting algorithm and you could add the "absorption peak scanning strategy".

**Response**: Our CRDS analysers just have a factory standard setting for absorption peak scanning strategy, so we did not mention it here.

P 11, L 9: Replace "checked" by "tested"

**Response**: I revised here (see p. 13, L. 36 in the marked manuscript).

P 11, L 12: Replace "confirmed" by "showed"

**Response**: I revised here according to your suggestion (see p. 13, L. 38 in the marked manuscript).

P 12, L 6: Aren't the lower RMSE values for the L2130 compared to the L1102 mainly due to the higher precision of the L2130?

**Response**: Yes, the lower RMSE values for the L2130i is mainly due to the higher precision of the L2130i compared to the L1102i. We revised this sentence as follows: "The lower RMSE values of the L2130i analyser are mainly due to the higher precision of the L2130i analyser compared to the L1102i analyser." (see p. 14, LL. 16-18 in the marked manuscript).

P 15, L 9: And in other tropical areas to make it more general?

**Response**: According to your suggestion, we replaced here as follows: "in the Amazon rainforest and other tropical areas" (see p. 18, L. 10 in the marked manuscript).

P 15, L13: Also here I would rather use "water vapour mixing ratio level" instead of moisture (moisture is more general and could also for instance imply liquid water).

**Response**: Here, we replaced "moisture conditions" with "$H_2O$ concentration levels" (see p. 18, L. 14 in the marked manuscript). In addition, in the next sentence we replaced "moisture content" with "$H_2O$ concentration" (see p. 18, LL. 15-16 in the marked manuscript).

[revised manuscript text omitted]

---

## Author Comment (AC4) · 4 Jan 2021

Please note: the original reviewer's comments are printed in black, while our response is printed in orange.

**Review of 'Characterising water vapour concentration dependence of commercial cavity ring-down spectrometers for continuous onsite atmospheric water vapour isotope measurements in the tropics' by Komiya *et al.*, submitted to AMT**

This manuscript presents a characterisation of concentration dependence of 2 commercial cavity ring-down spectrometers at 4 high water concentration levels between 21500 and 41000 ppm. The authors assessed the calibration results using 4 different calibration strategies, and for each calibration strategy using 5 different fitting functions. The authors concluded with identifying the most appropriate calibration strategy and fitting choice for their two specific instruments.

The relative new aspects of the study are the focus of the calibration at high water concentration levels (21500 to 41000 ppm) and the assessment of different calibration strategies (including fitting choices). The manuscript is relatively well written. Below I have detailed 2 major points, and some minor issues with suggestions.

**Response**: Thank you for your support and suggestions. We have revised our manuscript following your comments and suggestions. The revised parts are shown with track changes in the marked-up version of the manuscript. We also revised all the Figures.

**Major comments**

1. One key aspect of the study is to test the 5 calibration strategies. A general question would be: what motivates the authors to choose *these specific* calibration strategies in the first place? In other words, what is the purpose or hypothesis, advantage, and disadvantage behind each specific strategy? It would be more clear and helpful if the authors could briefly clarify these questions before presenting the methods and results.

**Response**: Regarding "to test the 5 calibration strategies", actually we tested the **4** calibration strategies Following your suggestion, we explained why we used the 4 calibration strategies in the first paragraph in section 2.3 as follows: "…The DI1 and DI2 calibration strategies used a single standard water (DI1 or DI2) to correct for [$H_2O$]-dependence. In contrast, the DI1-DI2*1Pair and DI1-2*2Pairs strategies used the two standard waters (DI1 and DI2) for calibrating [$H_2O$]-dependence because we considered that [$H_2O$]-dependence might change between the two standard waters according to recent studies (Bonne et al., 2019; Weng et al., 2020). In addition, the DI1-2*2Pairs strategy used more calibration data than the DI1-DI2*1Pair strategy to obtain more robust calibration fittings of [$H_2O$]-dependence." (see p. 5, LL. 20-25 in the marked manuscript).

2. The authors have described a custom-built calibration unit. The unit's working principle is similar to the commercially available (since 2013?) Picarro Standards Delivery Module (A0101). What is the specific motivation for the authors to build their own calibration unit? What main differences or advantages do the authors achieve?

**Response**: The Picarro officially guarantees that the operational $H_2O$ concentration range of A0101 is between 200 and 30,000 ppm, which does not cover our aimed $H_2O$ concentration over 30,000 ppm. In addition, according to discussion with Picarro's staff, there is no easy way to run A0101 off with the L1102i model. Based on these reasons, we built up the calibration unit for ourselves.

Based on these reasons, we revised the beginning of section 2.1 as follows: "A setup with a commercial vaporizer coupled with a standard delivery module (A0211 and SDM, A0101, respectively; Picarro Inc., Santa Clara, CA, USA), guarantees the delivery of standard water vapour samples up to 30,000 ppm of $H_2O$, which does not cover the $H_2O$ concentration range we are expecting for the Amazon rainforest. In addition, according to discussion with Picarro's technicians, there is no easy way to run an A0101 with the L1102i model.

Therefore, we built a calibration system to routinely and automatically conduct onsite-calibration of CRDS analysers (Fig. 1)." (see p. 3, LL. 18-21 in the marked manuscript).

In addition, due to the customization, we were able to achieve "The customized heating system and buffer reservoir enabled us to produce a high moisture stream of standard water vapour samples" (see p. 3, L. 41 – p. 4, L. 1 in the marked manuscript).

**Detailed comments**

Page 1, Line 12: 'isotope ratios' is confusing here since it normally refers to R and not delta-notation. It may be replaced with 'isotope compositions'. The term 'isotope ratios' has been used in many places in the manuscript, and all should be reconsidered or replaced.

**Response**: Following your suggestions, we replaced 'isotope ratios' with 'isotope compositions' throughout the manuscript.

P 2, L 17: 'models' replaced to 'model '

**Response**: Following your suggestion, we revised here (see p. 2, L. 26 in the marked manuscript).

P 2, L 26: To make the structure more clear, suggest to add 'In addition, ' before 'Moreira et al.'.

**Response**: We added 'In addition, ' before 'Moreira et al.' (see p. 2, L. 37 in the marked manuscript). Additionally, based on RC1 suggestion and the context we made a new paragraph from "However, $H_2O$ concentrations within the Amazon…" (see p. 2, LL. 35-42 in the marked manuscript).

P 2, L 37: To make the sentence more clear, suggest to remove 'expected based on past measurement at the ATTO site' or place it in a new sentence.

**Response**: Agreed. We removed it (see p. 3, LL. 6-7 in the marked manuscript).

P 3, L25: Maybe remove '(Fig. 1)'. This is already referred to in the beginning.

**Response**: Following your suggestions, we removed it (see p. 4, L. 7 in the marked manuscript).

P 3, L40: Is the purpose to remove 'salt compounds'? I would suppose that there is very low salt contents in the fresh water (if the original water is fresh water and not sea water, otherwise it should be stated). I think the purpose is to in general remove contaminants.

**Response**: Yes, the original water is fresh water. We replaced 'salt compounds' with 'contaminants' (see p. 4, L. 22 in the marked manuscript).

P4, L21: The statement is very strong here so that it requires proof. If it is/can not be proved, it is rather a hypothesis or speculation. Also, this statement is in contradicted to the statement in P9, L15. Maybe replace 'any' to 'to a large extent the' or something like this.

**Response**: It is difficult to clearly prove that the rinsing and drying steps prevented **any** residual memory effects. Therefore, we revised the text as follows: 'These rinsing and drying steps were introduced to minimize the residual memory effect' (see p. 5, LL. 1-2 in the marked manuscript).

P4, L26: Not really common in the literature to use "[H2O]" as abbreviation. "[H2O]" has appeared in many places to represent 'water concentration level'. I personally feel it hinders the reading flow. Maybe it is better to use 'WCL' or simply 'concentration'?

**Response**: Following your suggestion, we did not abbreviate concentration level (see p. 5, L. 7 in the marked manuscript), and revised the corresponding parts throughout the manuscript.

P4, L29: 'Thus, the isotopic deviation values at 21,500 ppm are set to 0.' Personally think this sentence is redundant and can be removed with no harm.

**Response**: We removed this sentence (see p. 5, LL. 10-11 in the marked manuscript).

P4, L29: '(Fig. 2)' can be removed here.

**Response**: We cannot find '(Fig. 2)' at L. 29 on p. 4 in the previous manuscript, but we assumed it to be placed at L. 42 on p. 4 in the previous manuscript. After the revision following the RC1 reviewer, we did not remove here but changed here with "(Fig. 2b)" to help readers to find the place with the assigned Figure 2's letter (b) (see p. 5, LL. 31 in the marked manuscript).

P5, L7: '(Fig. 3)'. It is better to assign each subplot with a letter, and refer the description to the exact subfigure, for example '(Fig. 3a)'. With this context, suggest to add letters for all the subplots in Fig 3, 4, 6, and 7, as you have done for other figures.

**Response**: According to your suggestion, we assigned a figure letter to each subplot in Fig. 3 (see p. 10 in the marked manuscript), and revised here as follows: '(Figs. 3i and 3j)' (see p. 5, L. 40 in the marked manuscript). In addition, we assigned a figure letter to each subplot in Figs. 4, 6 and 7 (see p. 12 and p. 16-17 in the marked manuscript).

P5, L16: 'For example,'

**Response**: Done (see p. 6, L. 8 in the marked manuscript).

P5, L16: Suggest to remove '=', also for the other occasions throughout the manuscript.

**Response**: Done.

P5, L26: Suggest to remove 'For instance,'.

**Response**: Since we explain how many pairs the DI1-2*2Pairs strategy formed at 28 h interval as an example, we left 'For instance' here (see p. 6, L. 20 in the marked manuscript).

P5, L35-38: The sentence is redundant. Suggest to change to 'For each calibration strategy, we also calculated RMSE value for each of the [$H_2O$]-dependence 2D or 3D fittings'.

**Response**: Done (see p. 6, LL. 29-30 in the marked manuscript).

P6, Table 1: Remove the bold title.

**Response**: Done (see p. 8, Table 1 in the marked manuscript).

P9, L12: '(1.1%).' to '(1.1 %)'.

**Response**: We revised here and also added space between '~1.5' and '%' in the end of this sentence (see p. 11, LL. 17-18 in the marked manuscript).

P9, L12: Suggest to remove '(> 40,000 ppm)'.

**Response**: We removed it (see p. 11, L. 21 in the marked manuscript).

P9, L125: Should be 'lower' precision? I can see the standard deviation becomes larger at high concentrations.

**Response**: Due to the revision in this section suggested by the RC1 reviewer, this sentence is no longer used.

P9, L126: Following above, should it be 'decrease'?

**Response**: We found "both analysers had the highest $\delta^{18}$O and $\delta^2$H measurement precision for DI1 at 29,000 ppm even though variability in $H_2O$ concentration measurement was higher at 29,000 ppm ($H_2O$-$\sigma$ = 578.6 ppm) than at 21,500 ppm ($H_2O$-$\sigma$ = 243.6)." (see p. 11, LL. 35-37 in the marked manuscript). Therefore, we revised "The increase in measurement precision of L2130i and L1102i with [$H_2O$], despite larger [$H_2O$] variability," in the previous manuscript as follows: "The high measurement precision of L2130i and L1102i even with the larger $H_2O$ concentration variability…" (see p. 11, LL. 37-38 in the marked manuscript).

P9, L133: Remove 'At all H 2 O concentration levels, and '.

**Response**: We removed it (see p. 11, L. 28 in the marked manuscript).

P9, L137: Again, the statement here sounds strong. It seems that it has been proved, but actually not in the manuscript. Therefore, I would say it is more like a speculation or hypothesis. Suggest to change 'mainly be' to 'is likely' or something similar.

Same applies to P11, L17.

**Response**: Following your suggestion, we revised the corresponding parts (see p. 11, L. 29 and p. 14, L. 2 in the marked manuscript).

P10, L8: Add space before unit '‰' and check all other units to be consistent.

**Response**: We revised the corresponding parts throughout the manuscript.

P10, L16: Should be 'Fig. 5a-d'.

**Response**: In this sentence, we mention the results only about the L2130i analyser. So, the Figure letters are correct, but to make it clear we added "L2130i's" before "…deviation of each isotope from reference" (see p. 13, L. 14 in the marked manuscript).

P10, L17: Should be 'Fig. 5b'.

**Response**: We revised here (see p. 13, L. 16 in the marked manuscript).

P10, L18-19: This is an interesting observation. There is recently also a systematic study on concentration dependence that emphasizes this isotope-dependency. Probably this is something worthy to be discussed?

*Weng, Y., Touzeau, A., and Sodemann, H.: Correcting the impact of the isotope composition on the mixing ratio dependency of water vapour isotope measurements with cavity ring-down spectrometers, Atmos. Meas. Tech., 13, 3167–3190, https://doi.org/10.5194/amt-13-3167-2020, 2020.*

**Response**: Following you suggestion, we revised here as follows: "This finding indicates that the L2130i's $\delta^2$H accuracy for high moisture like 41,000 ppm is dependent on the isotopic composition, such as has been found for low moisture conditions below 4,000 ppm (Weng et al., 2020). This further indicates that more than one

standard water needs to be used in the field under not only low moisture but also high moisture conditions." (see p. 13. LL. 16-21 in the marked manuscript).

P12, L10: Refer exactly to subfigures such as (Fig 6e, j).

**Response**: We revised here as follows: "(Fig. 6h)" (see p. 14, L. 22 in the marked manuscript).

P12, L23: It would be helpful to explain the method before instead of going directly to the results. It is not clear for the readers what have been calculated here and how they are calculated.

**Response**: Following your suggestion, we revised this sentence as follows: "Hence, for the DI1-2*2Pairs strategy we obtained a [$H_2O$]-dependence fitting method, which only obtained minimum RMSE values of $\delta^{18}O$ and $\delta^2H$, from each two calibration pairs at each interval and then calculated a contribution rate of the respective five [$H_2O$]-dependence fittings (i.e., linear, quadratic, cubic, quartic, linear surface fitting methods) to the total calibration pairs at each interval (Fig. 7)." (see p. 15, LL. 25-29 in the marked manuscript).

P12, L39: 'smaller sample numbers for calculating RMSE (28 h: n=43, 196 h: n=19; c.f., Fig. 2)'. It is confusing here because it seems to me the short-interval calibration has larger sample numbers ( n=43).

**Response**: We are sorry that the description in the previous manuscript was misleading. The sample number for obtaining each RMSE value corresponds to the sample number during the assigned measurement cycles. For example, a 28 h interval using the DI1-2*2Pairs strategy with ID-[3,4] & [7,8] used 8 samples from two measurement cycles (i.e., DI1: 5, DI2: 6) for calculating RMSE value of each fitting method. (8 samples = 4 samples × 2 measurement cycles). Therefore, we revised the information in the parentheses as follows: "(e.g., 28 h with ID-[3,4] & [7,8]: n=8 (4 samples × 2 measurement cycles, i.e., ID-5,6), 196 h with ID-[3,4] & [31,32]: n=104 (4 samples × 26 measurement cycles, i.e., ID-5-30); c.f., Fig. 2b)." (see p. 15, LL. 21-22 in the marked manuscript).

P12, L31-39: This paragraph jumps back to Fig. 6. Suggest to move it up after L19, before presenting Fig 7.

**Response**: Following your suggestion, we moved this paragraph before presenting Figure 7, and also divided this paragraph into two paragraphs to make the context clearer (see p. 15, LL. 8-22 in the marked manuscript).

P16, L2: Replace 'the CRDS' to 'CRDS'.

**Response**: Following your suggestions, we revised here (see p. 19, L. 5 in the marked manuscript).

Fig S1: delta²H instead of delta$^{18}$O in first line.

**Response**: We revised here (see p. 2, Fig. S1 caption's first line in the marked supplement).

[revised manuscript text omitted]

**(a)**

| | 7h | 7h | 7h | 7h | 7h | 7h | 7h | 7h | 7h | | 7h | 7h | 7h | 7h | 7h | 7h | 7h | |
| DI1 | DI2 | DI1 | DI2 | DI1 | DI2 | DI1 | DI2 | DI1 | DI2 | ··· | DI1 | DI2 | DI1 | DI2 | DI1 | DI2 | DI1 | DI2 |
| Cali. ID: | 1 | 2 | 3 | 4 | 5 | 6 | 7 | 8 | 9 | 10 ··· | 41 | 42 | 43 | 44 | 45 | 46 | 47 | 48 |

**(b)**

| Calibration strategy | Interval (h) | | | | | | | |
| | 14 | 21 | 28 | 35 | 42 | ~ | 196 | 203 |
|---|---|---|---|---|---|---|---|---|
| DI1 | [1,3] [3,5] [5,7] ~ [43,45] [45,47] (n = 23) | NA | [1,5] [3,7] [5,9] ~ [41,45] [43,47] (n = 22) | NA | [1,7] [3,9] [5,11] ~ [39,45] [41,47] (n = 21) | ~ | [1,29] [3,31] [5,33] ~ [17,45] [19,47] (n = 10) | NA |
| DI2 | [2,4] [4,6] [6,8] ~ [44,46] [46,48] (n = 23) | NA | [2,6] [4,8] [6,10] ~ [42,46] [44,48] (n = 22) | NA | [2,8] [4,10] [6,12] ~ [40,46] [42,48] (n = 21) | ~ | [2,30] [4,32] [6,34] ~ [18,46] [20,48] (n = 10) | NA |
| DI1-DI2*1Pair | NA | [1,4] [2,5] [3,6] ~ [44,47] [45,48] (n = 45) | NA | [1,6] [2,7] [3,8] ~ [42,47] [43,48] (n = 43) | NA | ~ | NA | [1,30] [2,31] [3,32] ~ [18,47] [19,48] (n = 19) |
| DI1-DI2*2Pairs | NA | NA | [1,2] & [5,6] [2,3] & [6,7] [3,4] & [7,8] ~ [42,43] & [46,47] [43,44] & [47,48] (n = 43) | NA | [1,2] & [7,8] [2,3] & [8,9] [3,4] & [9,10] ~ [40,41] & [46,47] [41,42] & [47,48] (n = 41) | ~ | [1,2] & [29,30] [2,3] & [30,31] [3,4] & [31,32] ~ [18,19] & [46,47] [19,20] & [47,48] (
[revised manuscript text omitted]